# On the exact entropy of $\mathcal{N}=2$ black holes

João Gomes, Huibert het Lam[a], Grégoire Mathys[b]

[a] *Institute for Theoretical Physics and Center for Extreme Matter and Emergent Phenomena, Utrecht University, Princetonplein 5, 3584 CE Utrecht, The Netherlands*
[b] *Institute for Theoretical Physics and Delta Institute for Theoretical Physics, University of Amsterdam, Science Park 904, 1098 XH Amsterdam, The Netherlands*

`joaomvg@gmail.com, h.hetlam@uu.nl, g.o.mathys@uva.nl`

**Abstract**

We study the exact entropy of four-dimensional $\mathcal{N}=2$ black holes in M-theory both from the brane and supergravity points of view. On the microscopic side the degeneracy is given by a Fourier coefficient of the elliptic genus of the dual two-dimensional $\mathcal{N}=(0,4)$ SCFT and can be extracted via a Rademacher expansion. We show how this expansion is mapped to a modified OSV formula derived by Denef and Moore. On the macroscopic side the degeneracy is computed by applying localization techniques to Sen's quantum entropy functional reducing it to a finite number of integrals. The measure for this finite dimensional integral is determined using a connection with Chern–Simons theory on $\mathrm{AdS}_2 \times \mathrm{S}^1$. The leading answer is a Bessel function in agreement with the microscopic answer. Other subleading corrections can be explained in terms of instanton contributions.

# 1 Introduction

Black hole entropy is an important tool to investigate the microscopic structure of a theory of quantum gravity such as string theory. At leading order the entropy is given by the Bekenstein–Hawking area law [1, 2] and since the seminal work of [3], this contribution to the entropy has been reproduced by counting microstates for black holes that preserve a sufficient amount of supersymmetry. The area law is universal as it follows from the Einstein–Hilbert action which appears in all consistent theories of gravity at very long distances. Hence, to really probe the microscopic structure of string theory, one has to take into account corrections to the leading order entropy contribution. In fact one can study the exact entropy which takes into account all perturbative and non-perturbative corrections to the leading area law.

In this paper we study the exact entropy of a class of four-dimensional $\mathcal{N} = 2$ extremal black holes in compactifications of M-theory on a Calabi–Yau threefold $CY_3$. More precisely, we consider the Maldacena, Strominger, Witten (MSW) setting [4], which has an M5-brane in an asymptotic geometry $\mathbb{R}^{1,3} \times S^1 \times CY_3$. The brane is wrapped on $S^1 \times \mathcal{P}$, where $\mathcal{P} \subset CY_3$ is a positive divisor. In addition we assume $\mathcal{P}$ to be irreducible, i.e. it cannot be written as the sum of two positive divisors. In the IR the worldvolume theory of this M5-brane flows to a two-dimensional $\mathcal{N} = (0, 4)$ superconformal field theory (SCFT). Microscopically, we assume that the exact degeneracy is given by the index [5] and thus by a Fourier coefficient of the elliptic genus of this SCFT [6–9]. Macroscopically, the number of microstates is given by the Sen quantum entropy functional [10, 11], which is a functional integral over string fields in the $AdS_2$ part of the near horizon geometry $AdS_2 \times S^2$ of the four-dimensional black hole. We are always interested in the microcanonical ensemble, which means considering fixed charges.

On the microscopic side, the modular properties of the elliptic genus can be used to write the degeneracy as a Rademacher expansion [12–14]. This is an infinite but convergent sum over Bessel functions each of which comes multiplied by a generalized Kloosterman sum. The series is completely fixed by the knowledge of the Fourier coefficients with negative polarity, i.e. the polar degeneracies. We determine explicit expressions for these Kloosterman sums, which to the best of our knowledge are new in the literature. This appears in section 2.

The leading term in the Rademacher expansion can be compared to a formula derived by Denef and Moore [15] which is a refinement of the OSV proposal [16]. The main idea of [15] is to use modular properties of the D4-D2-D0 partition function[1] to write it as a function of the polar degeneracies. Some of these, the so-called extreme polar degeneracies, were determined using the wall-crossing formula. The index is then equal to a multi-dimensional integral of the resulting partition function. In order to rewrite it in a form that can be compared to the Rademacher expansion, we use a specific contour prescription. The formula by Denef and Moore has a contribution due to worldsheet

---

[1]As shown in a series of papers [17–22], when the divisor $\mathcal{P}$ is reducible the modular properties used in [15] change. We discuss this further in section 4.

instantons which we a priori leave divergent. In the process of evaluating the integrals we use analyticity and convergence properties to determine a finite truncation of this instanton contribution. We compare the truncation with the physical considerations of [15] where similar restrictions on the possible instanton contributions have been determined. After bringing the D4-D2-D0 degeneracy into the appropriate form, we explicitly determine the mapping to the corresponding term in the sum coming from the elliptic genus. In this way we find expressions for the extreme polar degeneracies in the Rademacher expansion.

On the macroscopic side, the Sen functional can be computed using localization in supergravity [23–26], resulting in a finite dimensional integral. This integral is very similar to the formula of Denef and Moore, however the latter also has a measure factor. In [27] a method was proposed to compute this measure factor and was subsequently applied to $\mathcal{N} = 4$ and $\mathcal{N} = 8$ black holes in string theory. The main idea is to lift the four-dimensional theory on $\mathrm{AdS}_2 \times \mathrm{S}^2$ to a five-dimensional theory on $\mathrm{AdS}_2 \times \mathrm{S}^1 \times \mathrm{S}^2$. Reduction on $\mathrm{S}^2$ then leads to a Chern–Simons theory on $\mathrm{AdS}_2 \times \mathrm{S}^1$. The partition function of a Chern–Simons theory on a compact manifold can be computed in the limit of large levels using a saddle-point approximation [28]. For a non-compact manifold as $\mathrm{AdS}_2 \times \mathrm{S}^1$, one can show that the partition function has a similar expression in the same limit [27], and it can thus be compared with the localization result. Evaluating the finite-dimensional integral in the latter formula in the limit of large levels allows us to determine the measure. The main goal of section 3 is to apply this method to our setting. Temporarily ignoring the instanton contributions we find the measure of the D4-D2-D0 formula. We will see that, as in [27], the theory including the instanton contributions suggests a Chern–Simons computation using renormalized levels. This way we can determine the full measure. From macroscopic considerations we also find constraints on the possible instanton contributions. As it turns out, we reproduce the microscopic degeneracy that followed from the D4-D2-D0 partition function, which corresponds to the leading term in the Rademacher expansion.

The paper is organized as follows. In section 2 we begin with a review of the MSW $\mathcal{N} = (0,4)$ SCFT and its elliptic genus. Here we also present explicit expressions for the Kloosterman sums. After that we review the formula derived by Denef and Moore, which we subsequently rewrite to obtain an expression that we match to the leading term in the Rademacher expansion. From this matching, we can extract an expression for the extreme polar degeneracies. In section 3 we turn to the macroscopics, starting by briefly reviewing the localization procedure that yields a finite-dimensional integral for the exact degeneracy. We then compute the measure in order to obtain the first term of the Rademacher expansion. In section 4 we then end with a discussion on how to compute non-perturbative terms in the Rademacher expansion. We also comment on the generalization of the microscopic analysis conducted in this paper to the case where $\mathcal{P}$ is reducible. We relegate the derivation of the Kloosterman sums to the two appendices.

# 2 Microscopic degeneracy

In this section we consider the MSW $\mathcal{N} = (0,4)$ superconformal field theory (SCFT) [4] that arises from the low energy limit of an M5-brane wrapped on an irreducible, positive divisor $\mathcal{P} \subset \mathrm{CY}_3$ in an $\mathbb{R}^{1,3} \times \mathrm{S}^1 \times \mathrm{CY}_3$ geometry, where $\mathrm{CY}_3$ is a Calabi–Yau threefold. This SCFT is dual to string theory on $\mathrm{AdS}_3 \times \mathrm{S}^2 \times \mathrm{CY}_3$. Here $\mathrm{AdS}_3 \times \mathrm{S}^2$ is the near horizon geometry of the uplift of the $\mathcal{N} = 2$ four-dimensional black holes we consider. We are interested in the microstates of these black holes which should thus be captured by the SCFT.

We consider two different expressions for the microscopic degeneracy. The first one follows from the modified elliptic genus of the MSW SCFT and can be expressed through a Rademacher expansion. With this, the degeneracy of a non-polar state is given in terms of Bessel functions, Kloosterman sums and the degeneracies of the polar states. The second expression is derived in [15] and is a formula for the exact degeneracy of D4-D2-D0 bound states dual to the above described M-theory setting. We evaluate the integrals in this expression using a particular contour prescription in order to bring the degeneracy in the same form as a Rademacher expansion. We then derive the mapping between the two expressions and are able to determine the so-called extreme polar degeneracies. We also find explicit expressions for the Kloosterman sums in the Rademacher expansion.

We first review the MSW SCFT in section 2.1 and then in section 2.2 we discuss how to extract the degeneracy from the elliptic genus. In section 2.3 we continue by studying the degeneracy following from the D4-D2-D0 partition function. We subsequently map the latter to the leading order term of the Rademacher expansion in section 2.4 in order to determine the extreme polar degeneracies.

## 2.1 MSW SCFT

In this section we review[2] the MSW construction [4] of the two-dimensional $\mathcal{N} = (0,4)$ SCFT. This SCFT arises from an M5-brane wrapping $\mathrm{S}^1 \times \mathcal{P}$, where $\mathcal{P} = p^a \mathcal{D}_a$ is an irreducible 4-cycle with $\mathcal{D}_a$ forming a basis of 4-cycles in $H_4(\mathrm{CY}_3, \mathbb{Z})$. The cycle $\mathcal{P}$ must be very ample. In addition to the M5-brane there are M2-branes wrapping 2-cycles $\Sigma_a \in H_2(\mathrm{CY}_3, \mathbb{Z})$ which lead to five-dimensional particles with charges $q_a$. Later we will understand these M2-brane bound states as fluxes on the M5-brane. In addition the M5-brane carries $q_0$ units of momentum along the circle $\mathrm{S}^1$. By reducing the M5-brane worldvolume theory on $\mathcal{P}$ down to $\mathbb{R} \times \mathrm{S}^1$ we obtain, in the IR, an $\mathcal{N} = (0,4)$ SCFT. This theory corresponds to excitations of the zero modes of the M5-brane worldvolume fields.

The M5-brane worldvolume theory is given by the six-dimensional $(0,2)$ multiplet and consists of five real scalars, a chiral two-form $\beta$ with associated field strength $h$ and fermions. Three of the five scalars parametrize the position of the string in the three non-compact directions in $\mathbb{R}^3$ and give rise to three scalars in the $(0,4)$ SCFT which are

---

[2]An excellent and more detailed introduction to the MSW CFT and its elliptic genus can be found in [8, 29].

both left- and right-moving. The remaining two scalars parametrize the position of the cycle $\mathcal{P}$ in the Calabi–Yau threefold and give rise to [4]

$$d_p = \frac{P^3}{3} + \frac{c_2 \cdot P}{6} - 2 \tag{2.1}$$

left- and right-moving scalars in the two-dimensional SCFT. Here

$$
\begin{aligned}
P^3 &= \int_{\text{CY}_3} P \wedge P \wedge P = d_{abc} p^a p^b p^c \,, \\
d_{abc} &= \int_{\text{CY}_3} \omega_a \wedge \omega_b \wedge \omega_c \,, \\
c_2 \cdot P &= \int_{\text{CY}_3} P \wedge c_2(\text{CY}_3) \,,
\end{aligned}
\tag{2.2}
$$

where we used the Poincaré dual of $\mathcal{P}$: $P = p^a \omega_a$, where $\omega_a$ form a basis of $H^2(\text{CY}_3, \mathbb{Z})$ and are the Poincaré duals of $\mathcal{D}_a$. The chiral two-form gives rise to $b_2^-(\mathcal{P})$ left-moving and $b_2^+(\mathcal{P})$ right-moving scalars. These numbers can be computed [4] and are equal to

$$b_2^-(\mathcal{P}) = \frac{2}{3} P^3 + \frac{5}{6} c_2 \cdot P - 1 \,,$$

$$b_2^+(\mathcal{P}) = \frac{1}{3} P^3 + \frac{1}{6} c_2 \cdot P - 1 \,. \tag{2.3}$$

The fermions in six dimensions give rise to $b_2^+(\mathcal{P}) - 1$ right-moving (and no left-moving) complex fermions. In addition there are two complex right-moving fermions due to the goldstinos arising from the broken supersymmetry and so in total the number of right-moving real fermions is

$$N_R^F = \frac{2}{3} P^3 + \frac{1}{3} c_2 \cdot P \,, \tag{2.4}$$

which equals the number of right-moving bosons $N_R^B$ as required by supersymmetry. Using the previous results, one can compute the left- and right-moving central charges. They are given by

$$c_L = N_L^B + \frac{1}{2} N_L^F = P^3 + c_2 \cdot P \,,$$

$$c_R = N_R^B + \frac{1}{2} N_R^F = P^3 + \frac{1}{2} c_2 \cdot P \,. \tag{2.5}$$

For large $q_0$, Cardy's formula

$$S = 2\pi \sqrt{\frac{c_L q_0}{6}} \tag{2.6}$$

matches with the black hole entropy area formula [4].

One of the scalars arising in the reduction of the chiral two-form plays a special role. From the expansion of the flux $h = \mathrm{d}\beta$

$$h = \mathrm{d}\phi^a \wedge \alpha_a \,, \quad \alpha_a \in H^2(\mathcal{P}, \mathbb{Z}) \,, \tag{2.7}$$

we can consider the right-moving scalar $\varphi = d_{ab}p^a\phi^b$ corresponding to $\imath^*t$, the pullback of the Kähler form $t \in H^2(\mathrm{CY}_3, \mathbb{Z})$, where $\imath : \mathcal{P} \hookrightarrow X$ is the inclusion map. Together with the right-moving part $X^i$ of the three non-chiral scalar fields that parametrize the motion of the M5-brane on $\mathbb{R}^3$ and the four right-moving Goldstinos, it forms the center of mass $(0, 4)$ supermultiplet [30].

### 2.1.1 Charge lattice

The magnetic charges take values

$$p \in H_4(\mathrm{CY}_3, \mathbb{Z}) = H^2(\mathrm{CY}_3, \mathbb{Z}) \cong \imath^* H^2(\mathrm{CY}_3, \mathbb{Z}) \,, \tag{2.8}$$

where we used that the map

$$\imath^* : H^2(\mathrm{CY}_3, \mathbb{Z}) \to H^2(\mathcal{P}, \mathbb{Z}) \tag{2.9}$$

is injective because $\mathcal{P}$ is ample [4]. On $H^2(\mathcal{P}, \mathbb{Z})$ we have the following inner product between two-forms:

$$\beta_1 \cdot \beta_2 = \int_{\mathcal{P}} \beta_1 \wedge \beta_2 \,. \tag{2.10}$$

This naturally induces an inner product for the forms $\omega_a$ as

$$\omega_a \cdot \omega_b = \int_{\mathcal{P}} \imath^*\omega_a \wedge \imath^*\omega_b = \int_{\mathrm{CY}_3} P \wedge \omega_a \wedge \omega_b = d_{abc}p^c \equiv d_{ab} \,, \tag{2.11}$$

which establishes a lattice $\Lambda$ for the magnetic charges. From the Hodge index theorem it follows that this lattice has signature $(1, b_2(\mathrm{CY}_3) - 1)$, where the positive eigenvalue corresponds to the pullback of the Kähler form [4] and $b_2(\mathrm{CY}_3)$ is the second Betti number. The vector $p$ is a so-called characteristic vector of $\Lambda$, i.e.

$$k^2 + p \cdot k \in 2\mathbb{Z} \tag{2.12}$$

for all $k \in \Lambda$ [29].

The electric charges should take values in the dual lattice $\Lambda^*$ since we must have $q \cdot p \in \mathbb{Z}$. The quadratic form on $\Lambda^*$ is $d^{ab} = (d_{ab})^{-1}$ and takes values in $\mathbb{Q}$ because $d_{ab}$ is integer quantized. On the other hand, M2-brane charges can be generated by turning on fluxes on the M5-brane [4]. These arise from the WZ-coupling of the M5-brane field strength $h = \mathrm{d}\beta$ to the M-theory three-form $C_3$ via the term

$$\int_{\mathbb{R} \times \mathrm{S}^1 \times \mathcal{P}} h \wedge C_3 \,. \tag{2.13}$$

By taking fluxes of $h$ on $\mathrm{S}^1 \times \Sigma$ with $\Sigma \in H_2(\mathcal{P}, \mathbb{Z})$ we can generate M2-brane charges. Since $\beta \in H^2(\mathcal{P}, \mathbb{Z})$, one would naively think that the induced charges $q_i$ live in the lattice $H_2(\mathcal{P}, \mathbb{Z})$. However this contradicts the fact that $q \in H_2(\mathrm{CY}_3, \mathbb{Z})$. The solution to this puzzle is that the fluxes along two-forms $\beta_i$ which are not in $\imath^* H^2(\mathrm{CY}_3, \mathbb{Z})$ give

rise to charges that are conserved in the two-dimensional theory but are not conserved in the full M-theory [4, 30]. It actually turns out that the cancellation of the Freed-Witten anomaly [31] on the M2-brane worldvolume requires $q_a - d_{ab}p^b/2 \in \Lambda^*$ [32, 33].

For later use, we want to consider the projections $q_+$ and $q_-$ of a non-zero vector $q$ on respectively the positive and negative definite sublattices of $\Lambda^*$, such that $q_+^2 \geq 0$ and $q_-^2 \leq 0$. Since the positive direction of the lattice is given by $p^a$ we find that $q_{+a} = (q \cdot p/p^2)d_{ab}p^b$ and $q_{-a} = q_a - q_{+a}$.

### 2.1.2 Algebra and spectral flow

The symmetry algebra of the MSW $\mathcal{N} = (0,4)$ SCFT is a Wigner contraction of the large $\mathcal{N} = 4$ superconformal algebra [6, 34]. It consists of a small $\mathcal{N} = 4$ superconformal algebra together with four bosonic and four fermionic generators. They correspond to the fields in the center of mass multiplet, namely $(X^i, \varphi, \overline{\psi}_\alpha^\pm)$. The generators of the small $\mathcal{N} = 4$ algebra are four supercurrents $\overline{G}_\alpha^\pm$ with $\alpha = \pm$, three bosonic currents $\overline{J}^i$ that generate a level $k_R$ $SU(2)_R$ Kac–Moody algebra and the usual Virasoro generators with central charge $6k_R$. The momentum along the M-theory circle $q_0$ is given by the difference of the Virasoro generators as

$$q_0 = L_0 - \overline{L}_0 - \frac{c_L - c_R}{24}, \tag{2.14}$$

where $L_0$ is the Virasoro generator of the non-supersymmetric side. The other charges $q_a$ are eigenvalues of the $U(1)$ generators $J_a$. The small $\mathcal{N} = 4$ generators have non-trivial commutation relations with the center of mass fields. In particular, we have

$$\epsilon^{\alpha\beta}\{\overline{\psi}_{\alpha,0}^+, \overline{\psi}_{\beta,0}^-\} = \frac{2}{p^2},$$

$$\epsilon^{\alpha\beta}\{\overline{G}_{\alpha,0}^+, \overline{\psi}_{\beta,0}^-\} = 2\frac{q \cdot p}{p^2}, \tag{2.15}$$

$$\epsilon^{\alpha\beta}\{\overline{G}_{\alpha,0}^+, \overline{G}_{\beta,0}^-\} = 4\left(\overline{L}_0 - \frac{c_R}{24}\right).$$

It is straightforward to show using (2.15) that the usual BPS condition $\overline{G}_{\alpha,0}^\pm|BPS\rangle = 0$ needs to be modified. Indeed, for states that are not annihilated by all the supercharges, the correct condition is instead

$$\left(\overline{G}_{\alpha,0}^\pm - (q \cdot p)\overline{\psi}_{\alpha,0}^\pm\right)|BPS\rangle = 0. \tag{2.16}$$

Using (2.15), one can show that

$$\epsilon^{\alpha\beta}\left\{\overline{G}_{\alpha,0}^+ - (q \cdot p)\overline{\psi}_{\alpha,0}^+, \overline{G}_{\beta,0}^- - (q \cdot p)\overline{\psi}_{\beta,0}^-\right\} = 4\left(\overline{L}_0 - \frac{1}{2}\frac{(q \cdot p)^2}{p^2} - \frac{c_R}{24}\right). \tag{2.17}$$

Combining (2.16) and (2.17) implies

$$\overline{L}_0 - \frac{c_R}{24} = \frac{1}{2}\frac{(q \cdot p)^2}{p^2} = \frac{1}{2}q_+^2. \tag{2.18}$$

Note that the right-movers can carry momentum without breaking supersymmetry! In deriving (2.18) we have neglected the three-momentum $\vec{p}$ of the non-compact scalars.

The algebra is invariant under spectral flow that relates states with different charges:

$$J_{aL,0} \to J_{aL,0} + d_{ab}k_-^b \,,$$

$$L_0 \to L_0 - k_-^a J_{aL,0} - \frac{1}{2}k_-^2 \,. \tag{2.19}$$

There are similar expressions for the right-moving $\overline{L}_0$ and $J_{aR,0}$ operators. When $k \in \Lambda$, a state with charge $q_a \in \Lambda^* + p_a/2$ gets mapped to another state with charge in $\Lambda^* + p_a/2$:

$$q_a \to q_a + d_{ab}k^b \,,$$

$$q_0 \to q_0 - q \cdot k - \frac{1}{2}k^2 \,. \tag{2.20}$$

Hence spectral flow with $k \in \Lambda$ is a symmetry of the spectrum. Note that the quantity $\hat{q}_0 = q_0 + \frac{1}{2}q^2$ is left invariant by the spectral flow transformations (2.20).

## 2.2   Elliptic genus

Since the $\mathcal{N} = 2$ four-dimensional black holes preserve four supercharges, we need to count the half-BPS states of the (0,4) SCFT. This is done using the (modified) elliptic genus [6–8] given by

$$Z(\tau, \overline{\tau}, z) = \frac{1}{2}\mathrm{Tr}_{\mathrm{R}}(-1)^F F^2 e^{\pi i p \cdot q} e^{2\pi i \tau(L_0 - \frac{c_L}{24})} e^{-2\pi i \overline{\tau}(\overline{L}_0 - \frac{c_R}{24})} e^{2\pi i q \cdot z} \,, \tag{2.21}$$

where $F = 2\overline{J}_0^3$ is the fermion number. We introduced potentials $z^a$ for the $U(1)$ charges and the trace is taken in the Ramond sector. The double insertion of $F$ is needed because of the four fermionic zero modes [35, 36] and the factor $e^{\pi i p \cdot q}$ needs to be included since the charges take values in $\Lambda^* + p/2$ [8]. Note that if only right-moving ground states contributed, the elliptic genus would be a holomorphic function of $\tau$, but since we have states of the form (2.16) contributing, this is not the case. However, as we will see the non-holomorphic part is not problematic because it only contributes via theta functions. Neglecting the momentum modes on $\mathbb{R}^3$ we can use (2.14), (2.18) as well as the fact that the trace over the fermionic zero modes gives $\frac{1}{2}\mathrm{Tr}F^2(-1)^F = 1$ to find

$$Z(\tau, \overline{\tau}, z) = \sum_{q_0, q_a} d(q_0, q_a) e^{\pi i p \cdot q} e^{2\pi i \tau \hat{q}_0} e^{-\pi i \tau q_-^2} e^{-\pi i \overline{\tau} q_+^2} e^{2\pi i q \cdot z} \,. \tag{2.22}$$

Note that in our conventions $q_-^2 \le 0$ and $q_+^2 \ge 0$ and therefore the sum is convergent for $\mathrm{Im}(\tau) > 0$. From the spectral flow invariance it follows that the degeneracy $d(q_0, q_a)$ only depends on the value of $\hat{q}_0$ and the equivalence class of $q$ in $\Lambda^*/\Lambda + p/2$. Let us write $q = \mu + k + p/2$, where $k \in \Lambda$ and $\mu \in \Lambda^*/\Lambda$. Then from the spectral flow transformations (2.20) with $k \to -k$ we find that

$$d(q_0, q_a) = d\big(\hat{q}_0 - \tfrac{1}{2}(\mu + \tfrac{1}{2}p)^2, \mu + \tfrac{1}{2}p\big) \equiv d_\mu(\hat{q}_0) \,. \tag{2.23}$$

We can use (2.23) to rewrite the sum over $q_0$ and $q_a$ in (2.22) as a sum over $\hat{q}_0$, $\mu$ and $k$ in the following way:

$$Z(\tau, \overline{\tau}, z) = \sum_{\mu \in \Lambda^*/\Lambda} h_\mu(\tau) \Theta_\mu(\tau, \overline{\tau}, z) \,, \tag{2.24}$$

where the Narain–Siegel theta functions are defined as

$$\Theta_\mu(\tau, \overline{\tau}, z) = \sum_{k \in \Lambda} (-1)^{p \cdot (\mu + k + \frac{1}{2}p)} e^{-\pi i \tau (\mu + k + \frac{1}{2}p)_-^2} e^{-\pi i \overline{\tau}(\mu + k + \frac{1}{2}p)_+^2} e^{2\pi i (\mu + k + \frac{1}{2}p) \cdot z} \,. \tag{2.25}$$

The holomorphic functions $h_\mu(\tau)$ are given by

$$h_\mu(\tau) = \sum_{\hat{q}_0 \geq -c_L/24} d_\mu(\hat{q}_0) e^{2\pi i \tau \hat{q}_0} = \sum_{n=0}^\infty d_\mu(n) e^{2\pi i \tau (n - \Delta_\mu)} \,. \tag{2.26}$$

The lower bound on $\hat{q}_0$ is a result of supersymmetry [8] and

$$\Delta_\mu = \tfrac{1}{24}c_L - \left( \tfrac{1}{2}(\mu^2 + \mu \cdot p) - \lfloor \tfrac{1}{2}(\mu^2 + \mu \cdot p) \rfloor \right) \,, \tag{2.27}$$

where $\lfloor x \rfloor$ is the floor function, i.e. the greatest integer smaller than $x$. The second equality in (2.26) is found by substituting $q = \mu + k + p/2$ in $\hat{q}_0$ and using (2.12) together with

$$q_0 \equiv -\frac{p^2}{8} - \frac{c_L}{24} \pmod{\mathbb{Z}} \,, \tag{2.28}$$

which follows from the zero-point energies, see e.g. [37].

### 2.2.1 Modular transformation properties

The partition function transforms nicely under modular transformations

$$\gamma = \begin{pmatrix} a & b \\ c & d \end{pmatrix} \,, \qquad a, b, c, d \in \mathbb{Z} \,, \qquad ad - bc = 1 \,, \tag{2.29}$$

that act as

$$\tau \mapsto \frac{a\tau + b}{c\tau + d} \,, \qquad \overline{\tau} \mapsto \frac{a\overline{\tau} + b}{c\overline{\tau} + d} \,, \qquad z \mapsto \frac{z_-}{c\tau + d} + \frac{z_+}{c\overline{\tau} + d} \,, \tag{2.30}$$

where $z_-$ and $z_+$ are the projections on respectively the negative and positive definite part of $z$ as defined right before section 2.1.2. The modular group $PSL(2, \mathbb{Z})$ is generated by the transformations

$$T = \begin{pmatrix} 1 & 1 \\ 0 & 1 \end{pmatrix} \,, \qquad S = \begin{pmatrix} 0 & -1 \\ 1 & 0 \end{pmatrix} \,. \tag{2.31}$$

We denote its action on functions $f(\tau, \overline{\tau}, z)$ as

$$\mathcal{O}(\gamma) f(\tau, \overline{\tau}, z) \equiv f\left( \frac{a\tau + b}{c\tau + d}, \frac{a\overline{\tau} + b}{c\overline{\tau} + d}, \frac{z_-}{c\tau + d} + \frac{z_+}{c\overline{\tau} + d} \right) \,. \tag{2.32}$$

In order to derive the modular transformation properties of the elliptic genus, it is convenient to write it as [8]

$$Z(\tau, \overline{\tau}, z) = -\frac{1}{4\pi^2} \partial_v^2 W(\tau, \overline{\tau}, z, v)|_{v=\frac{1}{2}} , \qquad (2.33)$$

where we introduced the generalized partition function

$$W(\tau, \overline{\tau}, z, v) = \frac{1}{2} \text{Tr}_R \, e^{\pi i p \cdot q} e^{2\pi i \tau (L_0 - \frac{c_L}{24})} e^{-2\pi i \overline{\tau}(\overline{L}_0 - \frac{c_R}{24})} e^{2\pi i (q \cdot z + vF)} . \qquad (2.34)$$

The variable $v$ can be seen as a potential which implies that under modular transformations it transforms as

$$v \rightarrow \frac{v}{c\overline{\tau} + d} . \qquad (2.35)$$

We expect the partition function defined on a torus $W(\tau, \overline{\tau}, z, v)$ to be invariant under modular transformations. However, due to a global gravitational anomaly, it is actually only invariant up to a phase $\epsilon(\gamma)$. For $\gamma = T$ we find that

$$\mathcal{O}(T) W(\tau, \overline{\tau}, z, v) = e^{2\pi i (-\frac{p^2}{8} - \frac{c_L}{24})} W(\tau, \overline{\tau}, z, v) , \qquad (2.36)$$

where we have made use of (2.14) and (2.28). Then, using (2.5), the fact that we can write $p^2 = P^3$ due to (2.2) and (2.11), and the fact that $c_R/6$ is a Kac–Moody level and thus an integer, we find that

$$\epsilon(T) = e^{\pi i \frac{c_2 \cdot p}{12}} , \qquad (2.37)$$

where we have introduced the notation $c_2 \cdot p \equiv c_2 \cdot P$. The transformations $(ST)^3 = S^2 = I$ leave $\tau$ and $z$ invariant, which implies that

$$\epsilon(S) = \epsilon(T)^{-3} = e^{-\pi i \frac{c_2 \cdot p}{4}} . \qquad (2.38)$$

The most general phase $\epsilon(\gamma)$ then follows from (2.37) and (2.38). In deriving (2.22) we have neglected the momentum modes on $\mathbb{R}^3$. Including those modes in the right-hand side of (2.21) and extracting their contribution, one can use (2.33) to derive that [29]

$$\mathcal{O}(\gamma) Z(\tau, \overline{\tau}, z) = \epsilon(\gamma)(c\tau + d)^{-3/2}(c\overline{\tau} + d)^{1/2} e^{-\pi i \frac{cz_-^2}{c\tau + d}} e^{-\pi i \frac{cz_+^2}{c\overline{\tau} + d}} Z(\tau, \overline{\tau}, z) . \qquad (2.39)$$

The transformation of $h_\mu$ and $\Theta_\mu$ can be written as

$$\mathcal{O}(\gamma) h_\mu(\tau) \;=\; \epsilon(\gamma) (c\tau + d)^{-b_2/2 - 1} M(\gamma)^\nu{}_\mu h_\nu(\tau) , \qquad (2.40)$$

$$\mathcal{O}(\gamma) \Theta_\mu(\tau, \overline{\tau}, z) \;=\; (c\tau + d)^{b_2/2 - 1/2} (c\overline{\tau} + d)^{1/2} e^{-\pi i \frac{cz_-^2}{c\tau + d}} e^{-\pi i \frac{cz_+^2}{c\overline{\tau} + d}} M^{-1}(\gamma)^\nu{}_\mu \Theta_\nu (\tau, \overline{\tau}, z) ,$$

where $M(\gamma)$ form a representation of the modular group and $M(\gamma)^\nu{}_\mu M^{-1}(\gamma)^\rho{}_\mu = \delta^{\nu\rho}$. These transformation properties follow from the transformation of the partition function (2.39) and from the transformation properties of the theta functions, which we work out in the appendices A and B. This is done by determining the transformations for a much more general class of theta functions, generalizing the work of [38] (appendix A) and then

specifying to the theta functions we are interested in here (appendix B). The final result is given in (B.18). This way we find explicit expressions for the matrix components:

$$M^{-1}(\gamma)^\nu{}_\mu = \frac{1}{\sqrt{|\Delta|}} \frac{i\,(-i)^{b_2/2}}{c^{b_2/2}} \sum_{\substack{k\in\mathbb{Z}^n \\ k\,\mathrm{mod}\,c}} (-1)^{p\cdot(k+\mu-\nu)} e^{-\pi i\left[\frac{a}{c}(\mu+k+\frac{1}{2}p)^2 - \frac{2}{c}(\nu+\frac{1}{2}p)\cdot(\mu+k+\frac{1}{2}p) + \frac{d}{c}(\nu+\frac{1}{2}p)^2\right]},$$

(2.41)

where we introduced $\Delta = \det d_{ab}$. These explicit expressions have to the best of our knowledge not appeared in the literature previously. Their relation to generalized Kloosterman sums will become clear in the next section.

### 2.2.2  Rademacher expansion and its leading order term

From (2.40) we see that $h_\mu$ transforms with weight $-b_2/2 - 1$ under modular transformations. Since this number is negative, we can exploit the modular properties of $h_\mu(\tau)$ to find an exact expression for the degeneracies $d(q_0, q_a) = d_\mu(n)$ with $\hat{q}_0 = n - \Delta_\mu > 0$ using the Rademacher circle method [8, 12–14]. These are the degeneracies of the so-called non-polar states since they correspond to the non-polar terms of the function $h_\mu$. The Rademacher circle method gives[3]

$$d(q_0, q_a) = \sum_{m-\Delta_\nu<0} d_\nu(m) \sum_{c=1}^\infty \frac{i^{b_2/2+2}}{c^{b_2/2+3}} \sum_{\substack{-c\leq d<0 \\ (c,d)=1}} \epsilon(\gamma)^{-1} M^{-1}(\gamma)^\mu{}_\nu e^{2\pi i\left[\frac{a}{c}(m-\Delta_\nu)+\frac{d}{c}(n-\Delta_\mu)\right]}$$

$$\times \int_{\mathcal{C}} \mathrm{d}z\, z^{b_2/2+1} e^{2\pi i\left[(m-\Delta_\nu)\frac{i}{z} - (n-\Delta_\mu)\frac{iz}{c^2}\right]},$$

(2.42)

which we henceforth will refer to as the Rademacher expansion. Here $\gamma$ is of the form

$$\gamma = \begin{pmatrix} a & -\frac{1+al}{c} \\ c & -l \end{pmatrix} = \begin{pmatrix} a & b \\ c & d \end{pmatrix},$$

(2.43)

where $al \equiv -1 \pmod c$. The sum is over integers $c$, $d$ with greatest common divisor $(c, d) = 1$ and the integration is over the circle of radius $\frac{1}{2}$ and center $(\frac{1}{2}, 0)$ in the complex plane. The matrix coefficients $M^{-1}(\gamma)^\mu{}_\nu$ are given by (2.41). We finally note that the integral can be rewritten in terms of a modified Bessel function of the first kind. In addition

$$K(n - \Delta_\mu, m - \Delta_\nu, c) = i^{b_2/2+1} \sum_{\substack{-c\leq d<0 \\ (c,d)=1}} M^{-1}(\gamma)^\mu{}_\nu e^{2\pi i\left[\frac{a}{c}(m-\Delta_\nu)+\frac{d}{c}(n-\Delta_\mu)\right]}$$

(2.44)

is a generalized Kloosterman sum. We thus find that a degeneracy for $\hat{q}_0 > 0$ is expressed as a sum over Kloosterman sums, Bessel functions and the polar degeneracies, i.e. $d_\mu(\hat{q}_0)$ with $\hat{q}_0 < 0$.

---

[3]A very detailed derivation of this formula can be found in [29]. Formula (2.42) corresponds to (4.22) in this reference. However, we added the factor $\epsilon(\gamma)^{-1}$ that arises because of the (special) transformation (2.40) of $h_\mu$.

In section 2.4, we will map the $c = 1$ term in the Rademacher expansion (2.42) to the degeneracy coming from the D4-D2-D0 partition function. Using a saddle-point approximation of the Bessel functions one can see that the term corresponding to $c = 1$, $n = 0$ is the leading order contribution to the degeneracy in the large $\hat{q}_0$ limit. However, we will refer to the whole sum corresponding to $c = 1$ as the leading order term in the Rademacher expansion, even though there might be terms with $c > 1$ that dominate over particular terms with $c = 1$ (provided we do not take the large $\hat{q}_0$ limit). The choice of $c = 1$ implies that the only terms contributing in the sums over $d$ and $k$ (in (2.41)) are given by $d = -1$ and $k = 0$. Furthermore, we can choose $a = 0$. With these values the factor $\epsilon(\gamma)$ in (2.42) corresponds to the modular transformation

$$\gamma = \begin{pmatrix} 0 & -1 \\ 1 & -1 \end{pmatrix} = ST^{-1}, \tag{2.45}$$

such that from (2.37) and (2.38), it follows that

$$\epsilon(\gamma)^{-1} = e^{\pi i \frac{c_2 \cdot p}{3}}. \tag{2.46}$$

We can thus use (2.41) as well as (2.46) to rewrite the $c = 1$ term in (2.42) as

$$\begin{aligned}
d(q_0, q_a) &= \frac{-i}{\sqrt{|\Delta|}} e^{\pi i \frac{c_2 \cdot p}{3}} \sum_{m - \Delta_\nu < 0} d_\nu(m)(-1)^{p \cdot (\nu - \mu)} e^{2\pi i \left[ (\mu + \frac{1}{2}p) \cdot (\nu + \frac{1}{2}p) + \frac{1}{2}(\mu + \frac{1}{2}p)^2 \right]} e^{-2\pi i (n - \Delta_\mu)} \\
&\quad \times \int_{\mathcal{C}} dz \, z^{b_2/2+1} e^{-2\pi \left[ (m - \Delta_\nu) \frac{1}{z} - (n - \Delta_\mu)z \right]}.
\end{aligned} \tag{2.47}$$

We now rewrite this formula a bit for later convenience. From (2.27) it follows that

$$n - \Delta_\mu \equiv -\frac{c_L}{24} + \frac{1}{2}(\mu^2 + \mu \cdot p) \pmod{\mathbb{Z}}. \tag{2.48}$$

From (2.36) and (2.37) we then find that

$$e^{\pi i (\mu + \frac{1}{2}p)^2 - 2\pi i (n - \Delta_\mu)} = e^{-2\pi i \left( -\frac{p^2}{8} - \frac{c_L}{24} \right)} = e^{-\pi i \frac{c_2 \cdot p}{12}}. \tag{2.49}$$

Also, using the fact that we can write $q = \mu + k + p/2$ for $k \in \Lambda$, in follows that

$$e^{2\pi i (\mu + \frac{1}{2}p) \cdot (\nu + \frac{1}{2}p)} = e^{\pi i (\mu + \frac{1}{2}p) \cdot p} e^{2\pi i (q - k) \cdot \nu} = (-1)^{\mu \cdot p} e^{\frac{1}{2}\pi i p^2} e^{2\pi i q \cdot \nu}. \tag{2.50}$$

Combining the equalities (2.49) and (2.50) thus results in

$$\begin{aligned}
e^{\pi i \frac{c_2 \cdot p}{3}} (-1)^{p \cdot (\nu - \mu)} e^{2\pi i \left[ (\mu + \frac{1}{2}p) \cdot (\nu + \frac{1}{2}p) + \frac{1}{2}(\mu + \frac{1}{2}p)^2 \right]} e^{-2\pi i (n - \Delta_\mu)} &= (-1)^{p \cdot \nu} e^{2\pi i q \cdot \nu} e^{\pi i (\frac{1}{2}p^2 + \frac{c_2 \cdot p}{4})} \\
&= (-1)^{p \cdot \nu} e^{2\pi i q \cdot \nu} (-1)^{k_R}, \tag{2.51}
\end{aligned}$$

where we have identified the integer $k_R = c_R/6$. With this equality and the transformation $z \to 1/\tau$, we can rewrite the degeneracy (2.47) in the required form:

$$d(q_0, q_a) = \frac{i}{\sqrt{|\Delta|}} (-1)^{k_R} \sum_{m - \Delta_\nu < 0} d_\nu(m)(-1)^{p \cdot \nu} e^{2\pi i q \cdot \nu} \int_{\epsilon - i\infty}^{\epsilon + i\infty} \frac{d\tau}{\tau^{b_2/2+3}} e^{2\pi \frac{\hat{q}_0}{\tau} - 2\pi (m - \Delta_\nu)\tau}. \tag{2.52}$$

Here we changed the integration contour from $\epsilon = 1$ to an arbitrary $\epsilon > 0$. In section 2.4, by mapping (2.52) to the degeneracy coming from the D4-D2-D0 partition function, we find expressions for the polar degeneracies $d_\nu(m)$ corresponding to the so-called extreme polar states.

## 2.3 D4-D2-D0 partition function

The M-theory setting we consider can also be described using the dual type IIA picture, where it is a D4-D2-D0 bound state. This happens by taking the M-theory circle to be small. In this limit the M5-brane maps to a D4-brane, the M2-branes to D2-branes and the momentum maps to D0-brane charge. In [15], the authors derived a formula for the degeneracies of these bound states. This is done in the parameter regime where the background Kähler moduli take values $t_\infty^a = i\infty$. There, the D4-D2-D0 partition function can be expressed in terms of the polar degeneracies using its modular properties. The partition function is then truncated in such a way that the corresponding non-polar degeneracies only contain the $c = 1$ term of a Rademacher expansion. Using the wall-crossing formula, they subsequently determined the degeneracies of the so-called extreme polar states, i.e polar states that have a $\hat{q}_0{}^4$ satisfying

$$\eta \equiv \frac{\hat{q}_{0,\,\mathrm{min}} - \hat{q}_0}{\hat{q}_{0,\,\mathrm{min}}} \ll 1 \,. \tag{2.53}$$

It can be shown [15] that in the four-dimensional supergravity picture, these extreme polar states always correspond to a bound state of one D6- and one anti-D6-brane. Thus extreme polar states never correspond to a single black hole. Therefore, to determine the degeneracy of the bound state, one can move the moduli $t_\infty$ to a wall of marginal stability where the state splits in its constituents. The index must then factorize into the index of its constituents. The D6-D4-D2-D0 bound states with D6 charge $p^0 = 1$ can be counted by Donaldson–Thomas (DT) invariants. The Donaldson–Thomas partition function is then related to the topological string partition function. These properties can be used to determine the exact extreme polar state degeneracies by studying multiple decays. All of this results in a refined version of the OSV proposal [16] for the partition function. The corresponding non-polar degeneracies are given by

$$d(q_0, q_a) = \int \prod_{\Lambda=0}^{b_2} \mathrm{d}\phi^\Lambda \mu(p, \phi) e^{2\pi q_\Lambda \phi^\Lambda + \mathcal{F}(p,\phi)} \,, \tag{2.54}$$

where $\mathcal{F}(p, \phi)$ is a truncation of twice the real part of the free energy of the topological string and the integrals run over the imaginary $\phi$-axis. In principle, the function $\mathcal{F}$ is thus a function of the topological string coupling $g$ and the Kähler moduli $t^a$, but we make the identifications

$$g = \frac{2\pi}{\phi^0} \,, \qquad t^a = \frac{1}{\phi^0}(\phi^a + \frac{1}{2} ip^a) \,. \tag{2.55}$$

---

[4]To translate the expressions in [15] to our conventions one has to replace $q_0 \to -q_0$, $q_a \to -q_a$ and $\hat{q}_0 \to -\hat{q}_0$.

The measure $\mu(p, \phi)$ appearing in (2.54) is given in terms of $\mathcal{F}$ as [15]

$$\mu(p, \phi) = \frac{(-i)^{b_2}}{2\pi} \phi^0 \frac{\partial}{\partial \alpha} \mathcal{F}\big(g(\alpha), t(\alpha)\big)|_{\alpha=0} \,, \tag{2.56}$$

where

$$g(\alpha) = \frac{2\pi}{\phi^0} + 2\pi\alpha \,, \qquad t^a(\alpha) = \frac{1}{\phi^0}(\phi^a + \frac{1}{2}ip^a) + i\alpha p^a \,. \tag{2.57}$$

The function $\mathcal{F}$ can be decomposed into a perturbative and a non-perturbative part:

$$\mathcal{F}(g, t) = \mathcal{F}(g, t)_{\text{pert}} + \ln|Z_{\text{GW}}^\epsilon(g, t)|^2 \,. \tag{2.58}$$

Using that the perturbative part of the free energy is given by

$$F_{\text{pert}}^{\text{top}}(g, t) = -\frac{(2\pi i)^3}{6g^2} d_{abc} t^a t^b t^c - \frac{2\pi i}{24} c_{2a} t^a \,, \tag{2.59}$$

we find using (2.55) that

$$\mathcal{F}(p, \phi)_{\text{pert}} = 2\text{Re}\, F_{\text{pert}}^{\text{top}}(g, t) = \frac{2\pi}{24} \frac{p^2 + c_2 \cdot p}{\phi^0} - \frac{\pi}{\phi^0} d_{ab} \phi^a \phi^b \,. \tag{2.60}$$

For later convenience we note that when considering only the perturbative part $\mathcal{F} = \mathcal{F}_{\text{pert}}$, the measure (2.56) is

$$\mu(p, \phi)_{\text{pert}} = (-i)^{b_2} \phi^0 \left( \frac{p^2}{6} + \frac{c_2 \cdot p}{12} \right) \,. \tag{2.61}$$

The non-perturbative part $Z_{\text{GW}}^\epsilon(g, t)$ in (2.58) is a Gromov–Witten contribution due to worldsheet instantons. The superscript $\epsilon$ is used to indicate that this is a finite truncation of the full partition function $Z_{\text{GW}}(g, t)$, which we now consider first. The full partition function factorizes into the contribution of degenerate and non-degenerate instantons that are denoted by $Z_{\text{GW}}^0$ and $Z_{\text{GW}}'$ respectively:

$$Z_{\text{GW}}(g, t) = Z_{\text{GW}}^0(g) Z_{\text{GW}}'(g, t) \,. \tag{2.62}$$

The degenerate partition function has the product formula

$$Z_{\text{GW}}^0(g) = \left[ \prod_{n=1}^\infty (1 - e^{-gn})^n \right]^{-\chi(\text{CY}_3)/2} \,, \tag{2.63}$$

where $\chi(\text{CY}_3)$ is the Euler characteristic of the manifold. This partition function captures the contributions from worldsheets that wrap a point in the Calabi–Yau threefold. The non-degenerate partition function on the other hand captures the contributions from non-trivial maps of the worldsheet onto rational curves of arbitrary genus in the Calabi–Yau. This function can further be split into genus zero and non-zero genus contributions as

$$Z_{\text{GW}}'(g, t) = Z_{\text{GW}}^{h=0}(g, t) Z_{\text{GW}}^{h\neq 0}(g, t) \,, \tag{2.64}$$

where $h$ is the genus of the curve on which the worldsheet wraps. For genus 0 one gets the formula

$$Z_{\text{GW}}^{h=0}(g,t) = \prod_{n>0,m_a} \left(1 - e^{-g(\frac{1}{2}m\cdot p+n)}e^{2\pi im\cdot t}\right)^{nN_{q_a}}. \tag{2.65}$$

For non-zero genus the fluctuations can carry $SO(4) = SU(2)_L \times SU(2)_R$ quantum numbers $j_L, j_R$ [39–41]; $j_L$ is determined by the genus of the curve $C$ that the worldsheet wraps and $j_R$ is related to the weight of the Lefschetz action on the moduli space $\mathcal{M}_C$ of $C$. For genus zero the curve is isolated and therefore the fluctuations do not carry these internal quantum numbers. The partition function for non-zero genus is then given by

$$Z_{\text{GW}}^{h\neq 0}(g,t) = \prod_{n>0,m_L,m_a} \left(1 - e^{-g(\frac{1}{2}m\cdot p+n+2m_L)}e^{2\pi im\cdot t}\right)^{(-1)^{2j_R+2j_L}\cdot n(2j_R+1)N_{q_a,j_L,j_R}} \tag{2.66}$$

where $-j_L \leq m_L \leq j_L$ and $(j_L, j_R)$ are integral or half-integral. In (2.65) and (2.66) both the degeneracies $N_{q_a}$ and $N_{q_a,j_L,j_R}$ can be determined by a Schwinger type computation [39, 40].

We can expand the partition function $Z_{\text{GW}}$ as a series

$$Z_{\text{GW}}(g,t) = \sum_{(n,m_a)\in C} d(n,m_a)e^{-gn+2\pi im\cdot t}, \tag{2.67}$$

where $C$ is determined by the product formula for $Z_{\text{GW}}$. Note that $m \in \Lambda^*$ and that the partition function (2.67) is not convergent in general. In [15] it is truncated to $Z_{\text{GW}}^{\epsilon}(g,t)$ which is a finite sum. States in this truncation satisfy certain conditions such that the bound state of D6- and anti-D6-brane exists. These conditions can be found in the paper [15], but we come back to one of them in section 2.3.2. In this paper, we will also use the expansion (2.67), but without using the truncation of [15]. Instead we initially take[5]

$$C = \{(n, m_a) \in \mathbb{Z}^{b_2+1} | n, m_a \geq 0\} \tag{2.68}$$

and find additional restrictions on the ranges of $n$ and $m$ originating from analyticity and convergence properties. It is interesting to see whether we will obtain a similar truncation as in [15].

### 2.3.1 Recovering the Rademacher expansion

In this subsection, we want to rewrite (2.54) as a Rademacher expansion (2.42). Note that it is impossible to recover the complete sum since only terms with $c = 1$ and extreme polar degeneracies contribute to (2.54). Also we do not use the truncation of (2.67) that was considered in [15]. Therefore we can only reproduce part of (2.52). To recover the Bessel function structure, we use a specific contour prescription to explicitly compute the

---

[5]We take this domain for $n$ and $m$ to avoid polar terms in $Z_{\text{GW}}$. However, there might be also polar terms contributing to this partition function. For these terms one should do a similar analysis as we do for the non-polar terms in the rest of the paper.

integrals in (2.54). Doing this, we are left with a finite sum. The prescription is similar to the one originally used in [27, 42].

Using the results between (2.55) and (2.61) as well as the expansion (2.67), the degeneracy (2.54) becomes

$$d(q_0, q_a) = \sum_{(n_i, m_i) \in C} d(n_1, m_1) d(n_2, m_2) \mu(p, n_i, m_i) \int \prod_{\Lambda=0}^{b_2} d\phi^\Lambda \, \phi^0 e^{\mathcal{E}(p, \phi, n_i, m_i)} \,, \tag{2.69}$$

where the new measure $\mu$, that is independent of the potentials $\phi^\Lambda$, is given by

$$\mu(p, n_i, m_i) = (-i)^{b_2} \left[ \frac{p^2}{6} + \frac{c_2 \cdot p}{12} - (n_1 + n_2) - (m_1 + m_2) \cdot p \right]. \tag{2.70}$$

The exponent in (2.69) is equal to

$$\mathcal{E} = 2\pi q_\Lambda \phi^\Lambda + \frac{2\pi}{24} \frac{p^2 + c_2 \cdot p}{\phi^0} - \frac{\pi}{\phi^0} \phi^2 - \frac{2\pi}{\phi^0}(n_1 + n_2) + \frac{2\pi i}{\phi^0}(m_1 - m_2) \cdot \phi$$
$$- \frac{\pi}{\phi^0}(m_1 + m_2) \cdot p \,, \tag{2.71}$$

and it can be rewritten as

$$\mathcal{E} = \frac{2\pi}{\phi^0} \left[ \frac{c_L}{24} - (n_1 + n_2) - \frac{1}{2}(m_1 + m_2) \cdot p - \frac{1}{2}(m_1 - m_2)^2 \right] + 2\pi \hat{q}_0 \phi^0$$
$$- \frac{\pi}{\phi^0} \left[ \phi - \phi^0 q - i(m_1 - m_2) \right]^2 + 2\pi i q \cdot (m_1 - m_2) \,. \tag{2.72}$$

From this expression it is clear that the integral over $\phi$ in (2.69) is Gaussian. However, the matrix $d_{ab}$ has signature $(1, b_2 - 1)$. It is possible to modify the integration contour for $\phi$ in a way that makes this integral convergent. In addition, we choose it such that we recover the leading Bessel functions from the Rademacher expansion. The new integration contour is given as

$$\frac{1}{\phi^0} \equiv \tau = \epsilon + iy \,, \quad -\infty < y < \infty \,, \quad \epsilon > 0 \,. \tag{2.73}$$

Furthermore, we let

$$\phi_-^a = iu^a \,, \qquad \phi_+ = \frac{\varphi}{\tau} \,, \tag{2.74}$$

with $u, \varphi$ real.[6] The range of $u^a$ is not arbitrary and we have some limitations coming from requiring the Fourier expansion (2.67) to be convergent.[7] Writing the exponent in the expansion in terms of $y$, $u$ and $\varphi$ using (2.55) and the new contours, we find that we need

$$u \in U \equiv \{ u \in \mathbb{R} \mid -m \cdot p \leq 2(m \cdot u) \leq m \cdot p \quad \forall \, m_a \geq 0 \} \,. \tag{2.75}$$

---

[6]Note that here we denote the components of $\phi_-$ in the original basis and $\phi_+$ in the basis where $d_{ab}$ is split up in a positive definite and negative definite part.

[7]We do this by requiring the absence of polar terms in (2.67). See also footnote 5.

This way we also find that $\varphi$ can be arbitrary. Note that $u \in U$ is equivalent to $u$ satisfying $-p^a/2 \leq u^a \leq p^a/2$. Furthermore $u \cdot p = 0$ since $u$ is anti-self-dual and $p$ is self-dual. Combining these conditions it is easy to see that $U$ is non-empty.

We are now ready to obtain the final expression for the degeneracy. We can choose $\epsilon \gg 1$ in the contour (2.73) such that $|1/\phi^0| \gg 1$. Therefore, by transforming to a basis in which $d_{ab}$ is split up in a positive definite and a negative definite part, the Gaussian integral over $\phi_-$ can be computed via saddle-point approximation. That is,

$$\int \prod d\phi_- \, e^{-\frac{\pi}{\phi^0}\left[\phi-\phi^0 q-i(m_1-m_2)\right]_-^2} \simeq i^{b_2-1} \int \prod du \, e^{\frac{\pi}{\phi^0}[u-(m_1-m_2)_-]^2}$$

$$\simeq \begin{cases} i^{b_2-1}\dfrac{(\phi^0)^{(b_2-1)/2}}{\sqrt{\det(-d_{ab}^-)}} + \mathcal{O}\left(e^{\frac{\pi}{\phi^0}[u-(m_1-m_2)_-]_{\min}^2}\right), & (m_1-m_2)_- \in U\,, \\ \\ \mathcal{O}\left(e^{\frac{\pi}{\phi^0}[u-(m_1-m_2)_-]_{\max}^2}\right), & (m_1-m_2)_- \notin U\,, \end{cases} \tag{2.76}$$

where $\det(-d_{ab}^-)$ is the absolute value of the product of the negative eigenvalues of $d_{ab}$. We can also perform the integral over $\phi_+$ such that the leading contribution to the degeneracy becomes of Bessel type:

$$\begin{aligned} d(q_0, q_a) \;=\; & \frac{-i^{b_2-1}}{\sqrt{|\Delta|}} \sum_{\substack{(n_i,m_i)\in C \\ (m_1-m_2)_-\in U}} d(n_1, m_1)d(n_2, m_2)\mu(p, n_i, m_i) \\ & \times e^{2\pi i q \cdot (m_1-m_2)} \int_{\epsilon-i\infty}^{\epsilon+i\infty} \frac{d\tau}{\tau^{b_2/2+3}} e^{2\pi\frac{\hat{q}_0}{\tau}+2\pi\tau\mathcal{G}(p,n_i,m_i)} \end{aligned} \tag{2.77}$$

with

$$\mathcal{G}(p, n_i, m_i) = \frac{c_L}{24} - (n_1 + n_2) - \frac{1}{2}(m_1 + m_2) \cdot p - \frac{1}{2}(m_1 - m_2)^2\,. \tag{2.78}$$

### 2.3.2 Constraints on the possible instanton contributions

We now derive some conditions on the possible instanton contributions parametrized by the summation variables $n_1$, $n_2$, $m_1$, and $m_2$. We already have that $m_{ia}$, $n_i \geq 0$, and that $(m_1 - m_2)_- \in U$ in the sum in (2.77). However, there are additional constraints. First of all, when $\mathcal{G} \leq 0$ the contour in (2.77) can be closed such that there are no poles inside. This makes the integral vanish. In order to avoid that, we need to require

$$\mathcal{G}(p, n_i, m_i) > 0\,. \tag{2.79}$$

A second constraint can be derived by using the fact that vectors $u \in U$ satisfy

$$|2(m-n) \cdot u| \leq (m+n) \cdot p \tag{2.80}$$

for any $m, n \in \mathbb{Z}^{b_2}$ such that $m_a$, $n_a \geq 0$. In addition, using that $(m_1 - m_2)_- \in U$, and choosing $m = m_1$ and $n = m_2$, we obtain the following constraint:

$$-(m_1 - m_2)_-^2 \leq \tfrac{1}{2}(m_1 + m_2) \cdot p\,. \tag{2.81}$$

This readily implies the upper bound

$$\mathcal{G}(p, n_i, m_i) \leq \frac{c_L}{24} - (n_1 + n_2) - \frac{1}{4}(m_1 + m_2) \cdot p - \frac{1}{2}(m_1 - m_2)_+^2 \leq \frac{c_L}{24}. \qquad (2.82)$$

Physically, this indicates that the Gromov–Witten contributions are always subleading compared to the black hole saddle point.

This is as far as one can get by imposing both convergence and the fact that we want to obtain a Bessel function structure as in (2.77). However, as mentioned before, there are other constraints [15] that must be obeyed for a bound state of D6- and anti-D6-brane to exist. One of them ensures that the distance between the branes is positive and is, translated to our conventions, given by

$$\frac{p^2}{6} + \frac{c_2 \cdot p}{12} - (n_1 + n_2) - (m_1 + m_2) \cdot p > 0. \qquad (2.83)$$

We can try to derive (2.83) using the constraints that we have obtained so far. Unfortunately, we will not be able to show this in all generality, but we are able to show that the states for which this inequality is not satisfied are not extreme polar. First of all, from $\mathcal{G} > 0$ and (2.5) we find

$$\frac{c_2 \cdot p}{12} - (m_1 + m_2) \cdot p > -\frac{p^2}{12} + 2(n_1 + n_2) + (m_1 - m_2)_+^2 + (m_1 - m_2)_-^2. \qquad (2.84)$$

Therefore, for the left-hand side of (2.83), we find that

$$\frac{p^2}{6} + \frac{c_2 \cdot p}{12} - (n_1 + n_2) - (m_1 + m_2) \cdot p \; > \; \frac{p^2}{12} + (m_1 - m_2)_+^2 - \frac{1}{2}(m_1 + m_2) \cdot p, \qquad (2.85)$$

where we also used the restriction (2.81) and that $n_i \geq 0$. Writing $m_i = \alpha_i p + u_i$, where $p \cdot u_i = 0$ and $\alpha_i \geq 0$, we can specify when the right-hand side of (2.85) possibly fails to be positive. Namely, the inequality (2.85) becomes

$$\frac{p^2}{6} + \frac{c_2 \cdot p}{12} - (n_1 + n_2) - (m_1 + m_2) \cdot p > \left[ \frac{1}{12} + (\alpha_1 - \alpha_2)^2 - \frac{1}{2}(\alpha_1 + \alpha_2) \right] p^2. \qquad (2.86)$$

The right-hand side is positive provided

$$0 < \alpha_2 < \alpha_1 + \frac{1}{4} - \sqrt{\alpha_1 - \frac{1}{48}}, \qquad \alpha_2 > \alpha_1 + \frac{1}{4} + \sqrt{\alpha_1 - \frac{1}{48}}, \qquad (2.87)$$

in which case the inequality (2.83) is proven. This also means that we cannot show this inequality in full generality.

We will now use the result (2.92) of the next section in which we show that a state characterized by $n_i$ and $m_i$ has $\hat{q}_0 = -\mathcal{G}$. We can then examine the polarity (2.53) of a

state that has values $\alpha_1$, $\alpha_2$ which do not obey the bounds (2.87). In that case we have that

$$\frac{1}{12} + (\alpha_1 - \alpha_2)^2 - \frac{1}{2}(\alpha_1 + \alpha_2) < 0 \,. \tag{2.88}$$

In particular this translates into

$$\frac{1}{2}(m_1 + m_2) \cdot p + \frac{1}{2}(m_1 - m_2)_+^2 + \frac{1}{2}(m_1 - m_2)_-^2 \; > \; \frac{1}{4}(\alpha_1 + \alpha_2)p^2 + \frac{1}{2}(\alpha_1 - \alpha_2)^2 p^2$$

$$> \; \frac{1}{24}p^2 + (\alpha_1 - \alpha_2)^2 p^2 \,. \tag{2.89}$$

Hence:

$$\mathcal{G} < \frac{c_2 \cdot p}{24} - (n_1 + n_2) - (\alpha_1 - \alpha_2)^2 p^2 \leq \frac{c_2 \cdot p}{24} \ll \mathcal{G}_{\max} = \frac{p^2 + c_2 \cdot p}{24} \,, \tag{2.90}$$

where we have used that the M5-brane wraps a cycle $\mathcal{P}$ that is very ample, which implies that $p^2$ is very large. Hence the polarity (2.53) for states for which we cannot show (2.83) is almost one:

$$\eta \equiv \frac{\mathcal{G}_{\max} - \mathcal{G}}{\mathcal{G}_{\max}} > \frac{p^2}{p^2 + c_2 \cdot p} \,. \tag{2.91}$$

Therefore, the inequality (2.83) potentially fails for states that are far from extreme polar. This is not a problem since, as we already mentioned, the derivation in [15] is only valid for these extreme polar states.

## 2.4 Comparison between the D4-D2-D0 partition function and the elliptic genus

We can now finally compare the degeneracy (2.52), which is the leading order term in the Rademacher expansion, with the D4-D2-D0 degeneracy (2.77). This way we can determine expressions for the extreme polar degeneracies in (2.52).

From comparison of the integrals in (2.52) and (2.77) it is clear that one has to identify

$$\mathcal{G} = -m + \Delta_\nu \,. \tag{2.92}$$

Indeed we already found in (2.79) and (2.82) that $\mathcal{G}$ satisfies the same upper and lower bounds as $-m + \Delta_\nu$. Using the expressions (2.27) and (2.78), we find that the relation (2.92) becomes

$$-m - \tfrac{1}{2}(\nu^2 + \nu \cdot p) + \left\lfloor \tfrac{1}{2}(\nu^2 + \nu \cdot p) \right\rfloor = -(n_1 + n_2) - \tfrac{1}{2}(m_1 + m_2) \cdot p - \tfrac{1}{2}(m_1 - m_2)^2 \,. \tag{2.93}$$

This suggests the identifications

$$m \;=\; n_1 + n_2 + m_2 \cdot p + \left\lfloor \tfrac{1}{2}(m_1 - m_2)^2 + \tfrac{1}{2}(m_1 - m_2) \cdot p \right\rfloor \,,$$

$$\nu \;=\; m_1 - m_2 \,, \tag{2.94}$$

where we have used that $m_1 - m_2$ takes values in $\Lambda^*$ and the expression identified with $m$ is an integer. Using the upper bound (2.82) for $\mathcal{G}$ immediately implies that

$$n_1 + n_2 + m_2 \cdot p + \left\lfloor \tfrac{1}{2}(m_1 - m_2) \cdot p + \tfrac{1}{2}(m_1 - m_2)^2 \right\rfloor \geq 0\,, \tag{2.95}$$

which matches the non-negativity condition of $m$. It would also be very interesting to show whether the sum over $m_1$, $m_2$ contains more or less terms than the sum over $\nu$.

With the identifications above, the degeneracy (2.52) is equal to (2.77) where we have identified the polar degeneracies

$$d_\nu(m) \;\; = \;\; (-1)^{k_R + p \cdot \nu} \sum d(n_1, m_1) d(n_2, m_2) \left( \frac{p^2}{6} + \frac{c_2 \cdot p}{12} - (n_1 + n_2) - (m_1 + m_2) \cdot p \right). \tag{2.96}$$

The sum is over $(n_i, m_i) \in C$ such that $(m_1 - m_2)_- \in U$ and that (2.94) is satisfied. To derive (2.96) we have also used the definition (2.70) of the measure $\mu$. Note that we can trust (2.96) only for extreme polar states since only their degeneracies were taken into account in the derivation of (2.54). Also note that if one would have a different truncation of $Z_{\text{GW}}$, the identification (2.96) would be the same, however the summation domain would be different.

# 3 Macroscopic degeneracy

In the previous section, we reviewed how to obtain the degeneracies (2.54), before mapping them to the leading order term in the Rademacher expansion. In this section, we give a macroscopic derivation of (2.54). In section 3.1 we briefly review how the macroscopic entropy can be computed using Sen's quantum entropy functional [10, 11] as well as the localization technique [23, 24]. Localization reduces the functional to a finite number of integrals. This process also results in a measure which we determine under certain assumptions in section 3.2 using the approach developed in [27]. In section 2.3.2 we derived certain constraints on the possible instanton contributions to (2.54). We derive similar constraints from macroscopic considerations in section 3.3. We do not intend to give a macroscopic derivation of the full elliptic genus degeneracy, but only a partial derivation of the $c = 1$ term in the Rademacher expansion. We comment on extensions of this derivation in the discussion in section 4.

## 3.1 AdS$_2$ path integral via localization

Macroscopically the quantum degeneracy $d(q)$ of an extremal black hole with charges $q_\Lambda$ is proposed to be equal to [10, 11][8]

$$d(q_0, q_a) = \left\langle e^{-i \frac{q_\Lambda}{2} \oint_\theta A^\Lambda} \right\rangle_{\text{AdS}_2}^{\text{finite}}, \tag{3.1}$$

---

[8]The charges here are related to the ones in [10, 11] as $q_{\text{here}} = 2q_{\text{there}}$.

which is a functional integral over string fields in Euclidean $AdS_2$. It can be seen as an expectation value of a Wilson line inserted at the boundary. This functional is over all field configurations that asymptote to an $AdS_2$ Euclidean black hole with metric

$$ds^2 = \frac{\vartheta}{4}\left[(r^2-1)d\theta^2 + \frac{dr^2}{r^2-1}\right], \qquad A^\Lambda = -ie^\Lambda(r-1)d\theta, \qquad X^\Lambda = X_*^\Lambda, \qquad (3.2)$$

where $e^\Lambda$ and $X_*^\Lambda = \vartheta^{-1/2}(e^\Lambda + ip^\Lambda)$ are constants determined in terms of the charges by the attractor mechanism. In (3.2), the parameter $\vartheta$ is the size of $AdS_2$. These field configurations need to have fall-off conditions

$$ds^2 = \frac{\vartheta}{4}\left[(r^2+\mathcal{O}(1))d\theta^2 + \frac{dr^2}{r^2+\mathcal{O}(1)}\right], \quad A^\Lambda = -ie^\Lambda(r-\mathcal{O}(1))d\theta, \quad X^\Lambda = X_*^\Lambda + \mathcal{O}(1/r).$$
$$(3.3)$$

The superscript 'finite' in (3.1) refers to an $r = r_0$ cut-off prescription for regularizing and renormalizing the divergence caused by the infinite volume of $AdS_2$.[9] In the classical limit the degeneracy (3.1) becomes the exponential of the Wald entropy. To calculate the functional away from this limit we first integrate out the massive string and Kaluza–Klein modes such that we are left with a local Wilsonian effective action for the massless supergravity fields. The functional can then be computed using supersymmetric localization in $\mathcal{N} = 2$ off-shell supergravity [23, 24].

We first review how localization works for supersymmetric QFTs. Suppose there is a fermionic vector field $Q$ on a supermanifold $\mathcal{M}$ such that $Q^2 = H$ for some compact bosonic vector field $H$. Provided there is a fermionic function $V$ that is invariant under $H$, and that the measure $d\mu$, the action $S$ as well as the function $h$ are all invariant under $Q$, then the integral

$$\int_\mathcal{M} d\mu\, h\, e^{-S-\lambda QV} \qquad (3.4)$$

does not depend on $\lambda$. We can thus evaluate the integral for an arbitrary $\lambda$, and in particular we can choose $\lambda \to \infty$ such that we can use a saddle-point approximation. The integral thus localizes on the critical points of $QV$. In practice one can choose $V = (Q\Psi, \Psi)$, where $\Psi$ are the fermionic coordinates and ( , ) is a positive definite inner product defined on the fermions. Critical points of $QV$ are thus critical points of $Q$. This way we find that the integral (3.4) is equal to

$$\int_{\mathcal{M}_Q} d\mu_Q\, h\, e^{-S}, \qquad (3.5)$$

where $\mathcal{M}_Q$ is the submanifold of localizing solutions and $d\mu_Q$ is the induced measure on $\mathcal{M}_Q$. In supergravity, despite being more challenging, localization has been applied to compute the functional (3.1) [24–26]. In this case $\mathcal{M}$ is the field space of off-shell $\mathcal{N} = 2$ supergravity, $S$ the off-shell supergravity action with the right boundary terms, $h$ the Wilson line, $Q$ a certain supercharge and $\Psi$ all the fermionic fields of the theory. Since an

---

[9]Regularizations with more general cut-offs lead to the same renormalized action [5].

off-shell formulation of supergravity coupled to both vector- and hypermultiplets is not known, we shall ignore the contribution of hypermultiplets in the calculation. Considering $n_V + 1$ vector multiplets, where $n_V = b_2$ is the number of $\mathcal{N} = 2$ vector multiplets[10], coupled to the Weyl multiplet, the localizing solutions are given by [24, 43]

$$X^\Lambda = X_*^\Lambda + \frac{C^\Lambda}{r}, \qquad Y_1^{1\Lambda} = -Y_2^{2\Lambda} = \frac{2C^\Lambda}{r^2}, \qquad \hat{A} = -256\vartheta^{-1}, \tag{3.6}$$

where $\Lambda = 0, ..., n_V$, $X^\Lambda$ are the vector multiplet scalars, $Y^\Lambda$ and $\hat{A}$ denote auxiliary fields in the off-shell supergravity, $C^\Lambda$ are real constants and $\vartheta$ has been introduced in (3.2). All the other fields are fixed by their attractor values. The entropy functional (3.1) thus reduces to $b_2 + 1$ integrals over $C^\Lambda$ with as integrand the exponential of a renormalized action. Ignoring D-terms, this renormalized action evaluated on these solutions takes the form [24]

$$S_{\text{ren}} \equiv S + S_{\text{bdry}} - i\frac{q_\Lambda}{2} \oint A^\Lambda = 2\pi q_\Lambda \phi^\Lambda + \mathcal{F}(p, \phi), \tag{3.7}$$

where $\mathcal{F}$ is determined in terms of the prepotential $F(X, \hat{A})$:

$$\mathcal{F}(p, \phi) = 4\pi \operatorname{Im} F\left(\phi^\Lambda + \tfrac{1}{2}ip^\Lambda, -64\right). \tag{3.8}$$

In addition, we introduced the new scale invariant variables[11]

$$\phi^\Lambda = \frac{1}{2}(e^\Lambda + \sqrt{\vartheta}C^\Lambda). \tag{3.9}$$

Ignoring the D-terms in the calculation of (3.7) is partly justified by [44] in which many of these terms were shown to evaluate to zero. Using the (perturbative part of) the prepotential

$$F_{\text{pert}}(X, \hat{A}) = -\frac{1}{6}d_{abc}\frac{X^a X^b X^c}{X^0} - \frac{1}{24}c_{2a}\frac{X^a}{X^0}\frac{\hat{A}}{64}, \tag{3.10}$$

it follows that the perturbative part of (3.8) is equal to (2.60), where we used that $p^0 = 0$ since there are no D6-branes.

After localization, the quantum degeneracy takes the form [24]

$$d(q_0, q_a) = \int \prod_{\Lambda=0}^{b_2} d\phi^\Lambda \mathcal{M}(p, \phi)e^{2\pi q_\Lambda \phi^\Lambda + \mathcal{F}(p, \phi)}, \tag{3.11}$$

where $\mathcal{M}(p, \phi)$ is a measure we determine in the next section. The localization procedure does not give a contour prescription for the integrals over $\phi^\Lambda$. In sections 3.2.2 and 3.2.3 we choose the integration contours to be indefinite in a way that the integrals are convergent. To determine the measure in (3.11) the contribution from all the multiplets must, in principle, be taken into account. It has to be contrasted with the localization computation, where only the vector multiplets are taken into account. It is especially

---

[10]In the $\mathcal{N} = 2$ off-shell supergravity one needs to introduce a compensating vector multiplet.

[11]In [24] the gauge $\vartheta = 4$ was used and $\phi_{\text{here}} = \frac{1}{2}\phi_{\text{there}}$.

problematic to take into account the gravity multiplet in the localization computation. The localization technique depends on global supersymmetry and it is not a priori clear how to properly deal with it in supergravity. Progress in this direction has been made in [45, 46]. In the present work, we avoid these issues using an approach based on three-dimensional supergravity instead. For an approach using $\mathcal{N} = 2$ off-shell supergravity on AdS$_2\times$S$^2$ see [46–48].

## 3.2  Determining the measure

We can use the connection between AdS$_2$ and AdS$_3$ holography to determine the measure in (3.11) under certain assumptions which will become clear in the remainder of this section [27]. Namely, the four-dimensional type IIA geometry AdS$_2 \times$ S$^2$ corresponds after uplift to the five-dimensional M-theory geometry AdS$_2 \times$ S$^1 \times$ S$^2$, where S$^1$ is the M-theory circle[12]. Localization can be used to show that the five-dimensional entropy functional, which is computed on AdS$_2\times$S$^1\times$S$^2$, reduces to the four-dimensional version (3.11) [49, 50]. This equality only holds when the prepotential is given by its perturbative part. We can determine the measure using a saddle-point approximation (that we specify later) of the AdS$_2\times$S$^1$ partition function. Comparing the result with the same saddle-point approximation of (3.11) will enable us to read-off the measure $\mathcal{M}(p, \phi)$ [27].

The theory on AdS$_2 \times$ S$^1$ is obtained by reducing the five-dimensional theory on S$^2$, keeping only the massless fields. We need to impose AdS$_2$ boundary conditions (3.3) in order to make the connection with the two-dimensional case that we obtain by additionally reducing on S$^1$. It turns out that the measure corresponds to an anomaly of the Chern–Simons part of the three-dimensional theory. Namely, as we will see, it corresponds to the dependence of the one-loop determinant on the volume of the manifold. It is well known that Chern–Simons theory classically does not carry a dependence on the metric but when quantizing, one needs to pick a metric to ensure a well-defined gauge-fixed path integral [51]. Quantum mechanically the theory might depend on the volume of the manifold due to zero modes[13]. For the theory on AdS$_2$, only gauge fields can have normalizable zero modes which simplifies the analysis considerably, as we can focus on only the Chern–Simons terms in the three-dimensional theory [52, 53].

We are now ready to first approximate the AdS$_2\times$S$^1$ partition function, which we perform in section 3.2.1 and subsequently compare it to a saddle-point approximation of (3.11) in order to determine the measure. We do this first for the zero-instanton case in section 3.2.2 and then include the instanton contributions in section 3.2.3.

### 3.2.1  Chern–Simons partition function

The gauge field content of the theory on AdS$_2\times$S$^1$ can be easily determined as it corresponds to the current algebra in the dual SCFT which was discussed in section 2.1.2.

---

[12]In general, S$^1$ is actually fibered over AdS$_2$.

[13]In [51] the three-dimensional manifold is compact in contrast to the manifold here.

The small $\mathcal{N} = 4$ algebra contains four supercurrents $\overline{G}^{\pm\pm}$, three bosonic currents $\overline{J}^i$ that generate a level $k_R$ $SU(2)_R$ Kac–Moody algebra and the usual Virasoro generators. In addition we have $b_2$ $U(1)$ currents. The center of mass multiplet corresponds to one of these currents, namely the one along the direction with positive signature, and to three other bosonic and four fermionic currents. However, it has been argued before [54] that the center of mass modes are part of the black hole's hair. For this reason it seems they should not be included in the $AdS_2$ partition function which computes the contribution from horizon degrees of freedom of a single black hole. Because of this, we first focus on the other gauge fields while later commenting on the role of the center of mass modes. We thus focus on the dual description of the generators of the small $\mathcal{N} = 4$ algebra and the $b_2 - 1$ $U(1)$ generators corresponding to the directions with negative signature. This dual description is given by three-dimensional supergravity coupled to both an $SU(2)_R$ and $b_2 - 1$ abelian Chern–Simons terms. Since an Einstein–Hilbert action with negative cosmological constant in three dimensions can be written as a Chern–Simons action corresponding to $SL(2, \mathbb{R})_L \times SL(2, \mathbb{R})_R$ [55, 56], we can thus consider the Chern–Simons theory corresponding to the supergroup $SL(2, \mathbb{R})_L \times SU(1, 1|2)_R \times U(1)^{b_2-1}$.

We now need to compute the $AdS_2 \times S^1$ partition function using a saddle-point approximation. This can be done in the limit of large levels. When the Chern–Simons theory is defined on a compact manifold $\mathcal{M}_3$, the partition function for large level $r$ can be written as [27, 28]

$$Z(\mathcal{M}_3) \simeq \sum_A e^{2\pi i r \text{CS}(A)} \tau(\mathcal{M}_3, A)^{1/2} \big( r \, \text{vol}(\mathcal{M}_3) \big)^{(\dim H_A^1 - \dim H_A^0)/2}, \qquad (3.12)$$

where the sum is over gauge equivalence classes of connections $A$ with vanishing field strength, $\text{CS}(A)$ is the Chern–Simons action, $\tau^{1/2}$ the Reidemeister-Ray-Singer torsion and $H_A^0$, $H_A^1$ are the cohomology groups of the flat bundle. There can be zero modes for scalars and one-forms respectively when these groups are non-zero. We now assume that we can generalize (3.12) to the non-compact manifold $\mathcal{M}_3 = AdS_2 \times S^1$ and write the partition function as

$$Z \simeq \sum_A e^{2\pi i r \text{CS}(A)} Z_{\text{1-loop}}, \qquad (3.13)$$

where

$$Z_{\text{1-loop}} = \big( r \, \text{vol}(\mathcal{M}_3) \big)^{(\dim H_A^1 - \dim H_A^0)/2}, \qquad (3.14)$$

and we have neglected the torsion dependence because it is not known how to compute it for non-compact spaces. For our purposes of determining the measure it turns out that only the zero modes in the Chern–Simons theory are important, see also [27].

The metric on $AdS_2 \times S^1$ is given by

$$\text{d}s^2 = \frac{\vartheta}{4}\Big[ \sinh^2(\eta)\text{d}\theta^2 + \text{d}\eta^2 \Big] + \frac{\vartheta}{4(\phi^0)^2}\Big[ \text{d}y + i\phi^0(\cosh(\eta) - 1)\text{d}\theta \Big]^2, \qquad (3.15)$$

where we used the coordinate transformation $r = \cosh(\eta)$ with respect to (3.2) and $\theta$ and $y$ are periodically identified. Note that $S^1$ is actually fibered over $AdS_2$ but for convenience

we denote the space as a product. We now first consider the case where $\mathrm{vol}(\mathcal{M}_3) = 1/\phi^0$, with $1/\phi^0$ the radius of the circle $\mathrm{S}^1$. Recall that we use a cut-off prescription for regularizing and renormalizing the divergence caused by the infinite volume of $\mathrm{AdS}_2$, as we mentioned after (3.3). For a gauge field $A$ with level $r$ the $\mathrm{AdS}_2$ boundary conditions imply that $H_A^0$ vanishes [27]. However, the dimension of $H_A^1$ is infinite and needs to be regularized. This results in the following contribution to $Z_{\text{1-loop}}$ [27]:

$$\left(\frac{\phi^0}{r}\right)^{1/2}. \tag{3.16}$$

The gravitino can also have zero modes and it turns out that its contribution to $Z_{\text{1-loop}}$ cancels the contribution of the $SL(2,\mathbb{R})_R \times SU(2)_R$ gauge fields [27]. We are thus left with the contributions of $SL(2,\mathbb{R})_L$ and $b_2 - 1$ $U(1)$ gauge fields. Considering now the case where $\mathrm{vol}(\mathcal{M}_3) \neq 1/\phi^0$, and thus restoring the size $\vartheta$ of $\mathrm{AdS}_2$, gives an extra factor of $\vartheta$ to $Z_{\text{1-loop}}$ [27]. The one-loop result is therefore

$$Z_{\text{1-loop}} = \vartheta \sqrt{\frac{\phi^0}{k_L}} \prod_{i=1}^{b_2-1} \left(\frac{r_i}{\phi^0}\right)^{-1/2}, \tag{3.17}$$

where $k_L = c_L/6 = (p^2 + c_2 \cdot p)/6$ is the $SL(2,\mathbb{R})_L$ level [57]. At first, it is not clear what the value of the product $\prod_{i=1}^{b_2-1} r_i$ is. To investigate this, we need to revisit the center of mass modes. One of these modes corresponds to the $U(1)$ gauge field along the direction with positive signature. Including this extra gauge field, the product becomes

$$\prod_{i=1}^{b_2} r_i = \det d_{ab} = \Delta. \tag{3.18}$$

Inclusion of this abelian gauge field also seems natural from the reduction of M-theory on a Calabi–Yau threefold down to five dimensions, since this results in $b_2$ $U(1)$ gauge fields. This seems to indicate that the other fields in the center of mass multiplet also contribute. In addition, as we have seen in (2.15), this multiplet does not decouple from the rest of the $\mathcal{N} = 4$ generators. However, apart from the $U(1)$ gauge field, it is not clear how to capture the contribution of the center of mass multiplet fields in supergravity. We proceed by assuming that the same rule applies to each $U(1)$ gauge factor. After all, the one-loop contribution we are looking for only depends on zero modes of the gauge transformations and not on the specific details of the theory. Therefore we have to include $3+3$ additional $U(1)$ gauge fields dual to the holomorphic and anti-holomorphic currents $\partial_{z(\bar{z})} X^i$ and two additional gravitini fields. From the computation in [27], this adds a contribution

$$(\phi^0)^{(6-2)/2} = (\phi^0)^2 \tag{3.19}$$

to the one-loop partition function. The total one-loop result becomes

$$Z_{\text{1-loop}}^{\text{total}} = \frac{\vartheta}{\sqrt{k_L \Delta}} (\phi^0)^{b_2/2+5/2} = (-i)^{b_2-1} \frac{\vartheta}{\sqrt{k_L |\Delta|}} (\phi^0)^{b_2/2+5/2}, \tag{3.20}$$

where we used that $d_{ab}$ has $b_2 - 1$ negative eigenvalues which gives the exponent of $-i$. Note that (3.20) is valid for arbitrary values of $\phi^0$ since different values correspond to different background metrics.

### 3.2.2 Zero-instanton measure

We would like to compare (3.13), using the one-loop contribution (3.20), and (3.11) in the limit of large levels. We first do this in the zero-instanton sector where $\mathcal{F}(p, \phi) = \mathcal{F}(p, \phi)_{\text{pert}}$ and (3.11) is given by

$$d(q_0, q_a) = \int \prod_{\Lambda=0}^{b_2} \mathrm{d}\phi^\Lambda \mathcal{M}(p, \phi) e^{2\pi q_\Lambda \phi^\Lambda + \frac{2\pi}{24} \frac{p^2 + c_2 \cdot p}{\phi^0} - \frac{\pi}{\phi^0} \phi^2} . \tag{3.21}$$

We assume the measure does not contribute at the exponential level, and thus can be expanded polynomially around the attractor values. Performing the integrals over $\phi^a$ we get

$$d(q_0, q_a) \simeq \frac{1}{\sqrt{|\Delta|}} \int \mathrm{d}\phi^0 \, (\phi^0)^{b_2/2} \mathcal{M}(p, \phi^0, \phi^a_*) e^{2\pi \hat{q}_0 \phi^0 + \frac{\pi}{2} \frac{k_L}{\phi^0}} , \tag{3.22}$$

where $\phi^a_*$ denotes the the saddle point value. The exponent can be rewritten as

$$2\pi \hat{q}_0 \phi^0 + \frac{\pi}{2} \frac{k_L}{\phi^0} = \pi \sqrt{\hat{q}_0 k_L} \left( \frac{1}{x} + x \right), \qquad x \equiv \frac{1}{2\phi^0} \sqrt{\frac{k_L}{\hat{q}_0}} , \tag{3.23}$$

which implies that in the limit of large $k_L$ the integral localizes on $x \sim \pm 1$. We choose the positive value since $\phi^0$ is positive. Changing variables to $x$ and performing the method of steepest descent leads to

$$d(q_0, q_a) \simeq \frac{1}{\sqrt{k_L |\Delta|}} (\phi^0_*)^{b_2/2 + 3/2} \mathcal{M}(p, \phi^0_*, \phi^a_*) e^{2\pi \hat{q}_0 \phi^0_* + \frac{\pi}{2} \frac{k_L}{\phi^0_*}} , \tag{3.24}$$

where

$$\phi^0_* = \tfrac{1}{2} \sqrt{\frac{k_L}{\hat{q}_0}} \tag{3.25}$$

is the value at the saddle point. Note that this expression is valid for arbitrary values of $\phi^0_*$ and $\phi^a_*$ by tuning $\hat{q}_0$ and $q^a$ appropriately, provided we keep the value of $\hat{q}_0 k_L$ large. From (3.23) we see that the exponent in (3.24) can be rewritten as

$$2\pi \sqrt{\hat{q}_0 k_L} = \pi \frac{k_L}{\phi^0_*} , \tag{3.26}$$

which can be identified with the Chern–Simons action of flat $SL(2, \mathbb{R})_L \times SL(2, \mathbb{R})_R \times SU(2)_R$ connections [26][14]. The exponential in (3.24) thus corresponds to the exponential

---

[14]In [26] one has to take $a = d = 0$, $c = -b = 1$ and $\tau = i\phi^0$ to match our conventions. The exponent (3.26) then corresponds to (4.43) in this reference.

in (3.13) because flat abelian connections give vanishing Chern–Simons integrals. From comparison of (3.13) and (3.24) we find that the one-loop contribution is now given by

$$Z^{\text{total}}_{\text{1-loop}} = \frac{\mathcal{M}(p, \phi^0_*, \phi^a_*)}{\sqrt{k_L|\Delta|}}(\phi^0_*)^{b_2/2+3/2} \,. \tag{3.27}$$

Since both (3.20) and (3.27) are valid for arbitrary values of $\phi^\Lambda_*$, they should be equal. This enables us to determine the measure to be

$$\mathcal{M}(p, \phi) = (-i)^{b_2-1}\phi^0\vartheta \,. \tag{3.28}$$

The measure (3.28) is the main result of this section. However, we still need to determine a value for $\vartheta$. From comparing the degeneracy (3.11) together with this measure to the Denef and Moore formula (2.54) in the zero-instanton sector where the measure is given by (2.61), we find that this parameter is given by

$$\vartheta = \frac{p^2}{6} + \frac{c_2 \cdot p}{12} \,. \tag{3.29}$$

Unfortunately, we do not have a clear physical reason for this value. We will however provide $\vartheta$ with an interpretation. Because of the localization phenomena, the prepotential in (3.8) is evaluated in $X^\Lambda, \hat{A}$ values that correspond to the origin of $\text{AdS}_2$, i.e. $r = 1$ [24].[15] Therefore $\vartheta$ should be interpreted as the size of $\text{AdS}_2$ evaluated at the origin. Another comment we would like to make is that the value (3.29) corresponds to a choice of the factor in front of the Ricci scalar in the $\mathcal{N} = 2$ four-dimensional action. The Einstein–Hilbert term in this action is given by

$$\int \mathrm{d}^4x \, \sqrt{g} \, e^{-K(X,\overline{X},\hat{A})} R_g \,, \tag{3.30}$$

where

$$e^{-K(X,\overline{X},\hat{A})} = i(\overline{X}^\Lambda F_\Lambda - X^\Lambda \overline{F}_\Lambda) \,, \qquad F_\Lambda = \partial_\Lambda F(X, \hat{A}) \tag{3.31}$$

is the Kähler potential. We now evaluate the Kähler potential in $r = 1$. Using (3.6), (3.9) as well as the value for $X^\Lambda_*$ given after (3.2), we find that $\sqrt{\vartheta}X^\Lambda = 2(\phi^\Lambda + \frac{1}{2}ip^\Lambda)$ and $\vartheta\hat{A} = -256$. Using the zero-instanton prepotential (3.10) we then compute

$$e^{-K(X,\overline{X},\hat{A})} = \frac{4\vartheta^{-1}}{\phi^0}\left(\frac{p^2}{6} + \frac{c_2 \cdot p}{12}\right) \,. \tag{3.32}$$

The choice (3.29) is thus equivalent to choosing $4/\phi^0$ for this factor. It would be very interesting to understand this choice from a first principles calculation. However, loop calculations using Chern–Simons theories in non-compact spaces are not well-understood.

---

[15]Recall from (3.6) that in the origin $X^\Lambda = \vartheta^{-1/2}(e^\Lambda + ip^\Lambda) + C^\Lambda$ and $\hat{A} = -256\vartheta^{-1}$, where we also used the attractor values given below (3.2). The prepotential in (3.8) can be seen to be evaluated in these values by using that it is equal to $\frac{1}{4}\vartheta F\big(\vartheta^{-1/2}(2\phi^\Lambda + ip^\Lambda), -256\vartheta^{-1}\big)$ and then substituting the definition (3.9) of $\phi^\Lambda$.

### 3.2.3 Instanton contributions

We now consider instanton contributions to the AdS$_2$ path integral and argue that they give rise to the subleading Bessel functions. The microscopic answer (2.69) hints to additional saddle points in the gravity path integral because each Bessel function in (2.77) corresponds to a different saddle point for the scalars $\phi$. Each of these contributions is always exponentially subleading, and thus non-perturbative, in the limit of large electric charges and fixed magnetic charges. The exponents of the Bessel functions in (2.77) also strongly suggest that the prepotential gets renormalized by the instanton contributions. Without entering in details about this mechanism which has been extensively discussed in [58], we assume that for each instanton sector there is a renormalized prepotential that we determine below. This way we can read-off the effect of the instantons and interpret this in terms of an effective Chern–Simons theory which allows us to determine the measure originating from the instanton terms.

Let us assume that (3.11) is still valid for $\mathcal{F} = \mathcal{F}_{\text{pert}} + \mathcal{F}_{\text{non-pert}}$, where we determine $\mathcal{F}_{\text{non-pert}}$ using a non-perturbative prepotential. In analogy with section 2.3.1 we use a Fourier expansion for the non-perturbative part $\mathcal{F}_{\text{non-pert}}$:

$$e^{\mathcal{F}(p,\phi)_{\text{non-pert}}} = \sum_{(n_i,m_i)\in C} d(n_1,m_1)d(n_2,m_2)e^{-\frac{2\pi}{\phi^0}(n_1+n_2)+\frac{2\pi i}{\phi^0}(m_1-m_2)\cdot\phi-\frac{\pi}{\phi^0}(m_1+m_2)\cdot p} . \quad (3.33)$$

The degeneracy then becomes

$$d(q_0,q_a) = \sum_{(n_i,m_i)\in C} d(n_1,m_1)d(n_2,m_2)\int \prod_{\Lambda=0}^{b_2} d\phi^\Lambda \, \mathcal{M}(p,\phi,n_i,m_i)e^{\mathcal{E}(p,\phi,n_i,m_i)} , \quad (3.34)$$

where $\mathcal{E}$ is given by (2.71). As in [27, 58, 59] we now interpret this sum as the contribution of $d(n_1,m_1)$ instantons localized at the north pole of S$^2$ and $d(n_2,m_2)$ at the south pole. We thus indeed can interpret it as a sum over additional saddle points. We can now read-off the renormalized prepotential for each of these saddle points. Concretely, let us write $\mathcal{E} = 2\pi q_\Lambda \phi^\Lambda + \mathcal{F}$ as in the exponential in (3.11). Using (2.60) and (3.10) we then find that

$$\mathcal{F} = -2\pi i\left[F(\phi^\Lambda + \tfrac{1}{2}ip^\Lambda,-64,n_1,m_1) - \overline{F(\phi^\Lambda + \tfrac{1}{2}ip^\Lambda,-64,n_2,m_2)}\right], \quad (3.35)$$

with the renormalized prepotential

$$F(X,\hat{A},n,m) = F_{\text{pert}}(X,\hat{A}) + m_a\frac{X^a}{X^0}\frac{\hat{A}}{64} - \frac{i}{X^0}n\left(\frac{\hat{A}}{-64}\right)^{3/2} . \quad (3.36)$$

We would like to stress that for the moment the form of this prepotential is just a supposition and that we do not understand the physical details. For each of the instanton contributions in (3.34) we would like to determine the measure $\mathcal{M}(p,\phi,n_i,m_i)$. From section 2.3.1 we know that the function $\mathcal{E}$ can be rewritten as

$$\mathcal{E} = \frac{2\pi}{\phi^0}\left[\frac{c_L}{24} - (n_1+n_2) - \frac{1}{2}(m_1+m_2)\cdot p - \frac{1}{2}(m_1-m_2)^2\right] + 2\pi\hat{q}_0\phi^0$$

$$-\frac{\pi}{\phi^0}\left[\phi - \phi^0 q - i(m_1-m_2)\right]^2 + 2\pi i q\cdot(m_1-m_2) . \quad (3.37)$$

We then repeat the procedure of the previous section, and use saddle-point approxima-tions for the integrals over $\phi^a$ and $\phi^0$. This way we see that we get an expression as in (3.24), which has the form of the $SL(2)_L \times SL(2)_R \times SU(2)_R$ Chern–Simons action but now with a renormalized level

$$k_L = \frac{c_L}{6} - 4(n_1 + n_2) - 2(m_1 + m_2) \cdot p - 2(m_1 - m_2)^2 \,. \tag{3.38}$$

We also get an interesting extra phase given by the factor $2\pi i q \cdot (m_1 - m_2)$. From the microscopic point of view this corresponds to a Kloosterman phase as we have seen in section 2.2.2. From this interpretation we can derive the measure as in the zero-instanton sector. From the one-loop formula (3.20) we find that

$$\mathcal{M}(p, \phi, n_i, m_i) = (-i)^{b_2 - 1} \phi^0 \vartheta(p, n_i, m_i) \,, \tag{3.39}$$

where $\vartheta(p, n_i, m_i)$ is the quantum corrected $\text{AdS}_2$ size. In the previous subsection we have seen that our choice for $\vartheta$ in the zero-instanton sector is equivalent to requiring the Kähler potential to equal $4/\phi^0$ in the origin. Assuming the latter still holds when the instanton contributions are not neglected we can determine $\vartheta$. From the instanton corrected prepotential (3.36) we determine the Kähler potential by evaluating

$$e^{-K(X, \overline{X}, \hat{A}, n_i, m_i)} = i \left( \overline{X}^\Lambda F_\Lambda(X, \hat{A}, n_1, m_1) - X^\Lambda \overline{F_\Lambda(X, \hat{A}, n_2, m_2)} \right) \tag{3.40}$$

with $\sqrt{\vartheta} X^\Lambda = 2(\phi^\Lambda + \frac{1}{2} i p^\Lambda)$ and $\vartheta \hat{A} = -256$. In (3.40) the sections $F_\Lambda$ and $\overline{F}_\Lambda$ are no longer complex conjugate as this is not needed in Euclidean signature [60, 61]. We then evaluate (3.40) and find

$$e^{-K(X, \overline{X}, \hat{A}, n_i, m_i)} = \frac{4\vartheta^{-1}}{\phi^0} \left[ \frac{p^2}{6} + \frac{c_2 \cdot p}{12} - (n_1 + n_2) - (m_1 + m_2) \cdot p \right] \,. \tag{3.41}$$

Thus by requiring this to equal $4/\phi^0$ and using (3.39), we get the measure

$$\mathcal{M}(p, \phi, n_i, m_i) = (-i)^{b_2 - 1} \phi^0 \left[ \frac{p^2}{6} + \frac{c_2 \cdot p}{12} - (n_1 + n_2) - (m_1 + m_2) \cdot p \right] \,. \tag{3.42}$$

Up to a factor the degeneracy (3.34) with this measure matches the Denef and Moore formula (2.69) where the measure is given by (2.70). From the analysis in section 2.3.1 we then conclude that the $\text{AdS}_2$ partition function is a sum over Bessel functions with the same index but different arguments.

We can also interpret the quantum corrected prepotential (3.36) as a quantum cor-rection of the Kähler class. Namely, using topological string variables, (2.55) and (2.60), we can write (2.71) as $\mathcal{E} = 2\pi q_\Lambda \phi^\Lambda + \mathcal{F}$, where

$$\mathcal{F} = F^{\text{top}}(g, t, n_1, m_1) + \overline{F^{\text{top}}(g, t, n_2, m_2)} \,. \tag{3.43}$$

The renormalized free energy is defined as

$$F^{\text{top}}(g, t, n, m) = F^{\text{top}}_{\text{pert}}(g, t) + 2\pi i m \cdot t - gn \,. \tag{3.44}$$

The zero-instanton free energy (2.59) is just the integral of the Kähler class $t$ over the Calabi–Yau manifold:

$$F_{\text{pert}}^{\text{top}}(g,t) = -\frac{(2\pi i)^3}{6g^2}\int_{\text{CY}_3} t \wedge t \wedge t - \frac{2\pi i}{24}\int_{\text{CY}_3} t \wedge c_2(\text{CY}_3) \tag{3.45}$$

with $t = t^a \omega_a$ where $\omega_a \in H^{1,1}(\text{CY}_3)$. We now consider a quantum correction of the form

$$t \to t + \frac{g}{4\pi^2}G \tag{3.46}$$

with $\int_{\Sigma^a} G = 0$ for all two-cycles $\Sigma^a \subset H_{1,1}(\text{CY}_3)$ such that the volume of 2-cycles is preserved which ensures that we are still in the microcanonical ensemble. With this correction the free energy gets shifted to

$$F_{\text{pert}}^{\text{top}}(g,t) \to F_{\text{pert}}^{\text{top}}(g,t) + \frac{2\pi i}{2(2\pi)^2}\int_{\text{CY}_3} G \wedge G \wedge t + \frac{ig}{6(2\pi)^3}\int_{\text{CY}_3} G \wedge G \wedge G. \tag{3.47}$$

Suppose now that $G$ corresponds to a bundle with Chern numbers

$$-\frac{1}{2(2\pi)^2}\int_{\mathcal{D}^a} G \wedge G = -m_a \in \mathbb{Z}, \tag{3.48}$$

where $\mathcal{D}^a$ is the four-cycle Poincaré dual to $\omega_a$, and

$$-\frac{i}{6(2\pi)^3}\int_{\text{CY}_3} G \wedge G \wedge G = n \in \mathbb{Z}. \tag{3.49}$$

The shift (3.47) can then be rewritten as

$$F_{\text{pert}}^{\text{top}}(g,t) \to F_{\text{pert}}^{\text{top}}(g,t) + 2\pi i m \cdot t - gn, \tag{3.50}$$

which is precisely the quantum corrected free energy (3.44).

## 3.3 Constraints on the possible instanton contributions

From the microscopic derivation we know that there are certain constraints on the possible instanton contributions parametrized by $n_i$ and $m_i$ such that the sum over Bessel functions becomes finite. It is interesting to see which of these conditions derived in section 2.3.2 can be reproduced from the macroscopic description.

First of all, since the size $\vartheta$ is a positive quantity we need

$$\frac{p^2}{6} + \frac{c_2 \cdot p}{12} - (n_1 + n_2) - (m_1 + m_2) \cdot p > 0. \tag{3.51}$$

This is precisely the condition (2.83) derived by Denef and Moore. We could however not completely derive this inequality on the microscopic side by just following analyticity and convergence properties.

We also want the instantons in the sum (3.34) with larger topological charges to be less important in order to guarantee that we obtain the Cardy formula in the appropriate limit. This imposes a few other constraints that we can derive from (3.43) and (3.44). For the first constraint we use that $g = 2\pi/\phi^0$ and $n_i \geq 0$ such that the on-shell value of $\phi^0$ needs to be positive. From (3.25) and (3.38) this implies that

$$\mathcal{G}(p, n_i, m_i) = \frac{c_L}{24} - (n_1 + n_2) - \frac{1}{2}(m_1 + m_2) \cdot p - \frac{1}{2}(m_1 - m_2)^2 > 0 \,, \qquad (3.52)$$

since $\hat{q}_0 > 0$. This condition is the same as (2.79). A second condition follows from the dependence on $m_i$. Namely, using (3.44), we find that the on-shell values $t_* = \phi_* + \frac{1}{2}ip$ and $\bar{t}_* = \phi_* - \frac{1}{2}ip$ have to satisfy

$$\mathrm{Re}(im_1 \cdot t_*) \leq 0 \,, \qquad \mathrm{Re}(im_2 \cdot \bar{t}_*) \geq 0 \,. \qquad (3.53)$$

The on-shell value $\phi_*^a$ requires a bit of attention. As in section 2.3.1 we have to Wick rotate the anti-self-dual part of $\phi^a$ because its Gaussian has a wrong sign. From (3.37) we then find that

$$\phi_*^a = \phi_*^0 q^a + i(m_1 - m_2)^a \,, \qquad (3.54)$$

such that the conditions (3.53) become

$$m_1 \cdot (\tfrac{1}{2}p + m_1 - m_2) \geq 0 \,, \qquad -m_2 \cdot (-\tfrac{1}{2}p + m_1 - m_2) \geq 0 \,. \qquad (3.55)$$

Adding these two inequalities results in

$$-\tfrac{1}{2}(m_1 - m_2)^2 \leq \tfrac{1}{4}(m_1 + m_2) \cdot p \,, \qquad (3.56)$$

which immediately implies that

$$\mathcal{G}(p, n_i, m_i) \leq \frac{c_L}{24} - (n_1 + n_2) - \frac{1}{4}(m_1 + m_2) \cdot p \leq \frac{c_L}{24} \,. \qquad (3.57)$$

For the last inequality sign we used that $m_i \cdot p \geq 0$. The inequality (3.57) is the same as (2.82) on the microscopic side. The inequalities that we derived on the way are a bit stronger than in section 2.3.2. Here we see from (3.55) that $|(m_1 - m_2)^a| \leq p^a/2$, while we saw that on the microscopic side we only have that $|(m_1 - m_2)_-^a| \leq p^a/2$ which led to (2.81) instead of (3.56).

# 4 Discussion and outlook

We have studied the exact entropy of four-dimensional $\mathcal{N} = 2$ extremal black holes in M-theory. The black holes arise from an M5-brane wrapped on a circle and an irreducible divisor $\mathcal{P} \subset \mathrm{CY}_3$ in an asymptotic geometry $\mathbb{R}^{1,3} \times \mathrm{S}^1 \times \mathrm{CY}_3$. On the microscopic side, we have studied the degeneracy that follows from the elliptic genus of the two-dimensional (0,4) SCFT dual to these black holes. This degeneracy can be written as a Rademacher

expansion (2.42), i.e. a sum of Bessel functions dressed with generalized Kloosterman sums. We derived explicit expressions for these Kloosterman sums in section 2.2.1. In addition, we have mapped the Rademacher expansion to the degeneracy as derived by Denef and Moore [15]. On the macroscopic side, we derived the measure that arises in the localization computation [24] of the exact degeneracy given by the Sen quantum entropy functional [10, 11]. For this derivation we had to make a few assumptions and it would be interesting to study these better.

A natural extension of this work would be to study the case where the divisor $\mathcal{P}$ is reducible. The index then also counts the degeneracies of multi-centered configurations with the same asymptotic charges as the single-centered black hole [15, 62].[16] As shown in [17–22] the generating functions for the degeneracies in the case of reducible $\mathcal{P}$ are given by vector valued mock modular forms instead of the vector valued modular forms that one finds for irreducible $\mathcal{P}$. Mock modular forms also show up in compactifications with $\mathcal{N} = 4$ supersymmetry. In such compactifications the index is only sensitive to single-centered and two-centered black holes [64]. The meromorphic Jacobi form that counts these quarter-BPS states can be decomposed as a sum of a mixed mock Jacobi form and an Appell–Lerch sum [65]. The degeneracies of the single-centered black hole are then given by the Fourier coefficients of this mixed mock Jacobi form. Due to the mock modular behavior, the microscopic degeneracy of a single black hole is not given by a Rademacher expansion. However, for the $\mathcal{N} = 4$ case the modular properties can still be used to derive an asymptotic expansion for the Fourier coefficients of the mixed mock Jacobi form relevant for the counting of BPS states [66, 67]. The degeneracies are still given by a Rademacher-like expansion. The results of [67] might also be applied to the mixed mock modular forms that arise in the $\mathcal{N} = 2$ compactifications.

The microscopic degeneracy also includes degrees of freedom living outside of the horizon [68, 69]. We have indeed seen that at the macroscopic side we need to take into account the contribution of gauge fields corresponding to the center of mass multiplet to reproduce the localization measure, even though it is not clear how these contributions are captured by the supergravity theory. It is also puzzling that these degrees of freedom are interacting with the horizon degrees of freedom as the corresponding algebras do not commute.

At the macroscopic side we have only reproduced the leading Bessel function in (2.42) and given a heuristic explanation for some of the subleading Bessels. It would be useful to clarify the origin of the instanton contributions as a new family of saddles contributing to the partition function. This has been done for one-quarter BPS black holes in $\mathcal{N} = 4$ supergravity in [58]. The sums over $c$ and $d$ in the Rademacher expansion can be explained by including $(\mathrm{AdS}_2 \times \mathrm{S}^1 \times \mathrm{S}^2)/\mathbb{Z}_c$ orbifold geometries in the path integral [70] that satisfy the $\mathrm{AdS}_2$ boundary conditions. This has been done in [26] for $\mathcal{N} = 8$

---

[16]Also for irreducible $\mathcal{P}$ there might be states contributing to the index that have some substructure. As we have stated in section 2.3 extreme polar states always correspond to a bound state of one D6- and one anti-D6-brane. In [63] the contributing multi-centered configurations that build the polar sector have been identified in certain explicit settings.

compactifications and in [71] for $\mathcal{N} = 4$ compactifications. In particular this reproduces the Kloosterman sums from macroscopics. Our exact formulas for the Kloosterman sums can give similar insights into the gravity calculation, including the study of arithmetic properties and duality as has been done for $\mathcal{N} = 4$ and $\mathcal{N} = 8$ compactifications in [72].

**Acknowledgements**  We thank Alexandre Belin, Jan de Boer, Christopher Couzens, Nava Gaddam, Jan Manschot, Kilian Mayer, John Stout, Stefan Vandoren and Erik Verlinde for valuable discussions and correspondence.

H.L. is supported in part by the D-ITP consortium, a program of the Netherlands Organization for Scientific Research (NWO) that is funded by the Dutch Ministry of Education, Culture and Science (OCW), and by the NWO Graduate Programme. G.M. is supported in part by the ERC Starting Grang GENGEOHOL.

# A  Modular transformation general theta functions

In this appendix we derive how a theta function of the form

$$\Theta_{g,h}(\tau, \overline{\tau}, z; \alpha, N) = \sum_{m \equiv \alpha \, (N\Delta)} (-1)^{\frac{1}{N\Delta^2} h \cdot (m - \alpha)} e^{-\pi i \tau \frac{1}{N\Delta^2}(m + \frac{1}{2}g)_-^2} e^{-\pi i \overline{\tau} \frac{1}{N\Delta^2}(m + \frac{1}{2}g)_+^2} e^{\frac{2\pi i}{\Delta}(m + \frac{1}{2}g) \cdot z} \,, \tag{A.1}$$

transforms under a general element of the modular group. The function (A.1) is defined for $\tau$ in the upper half of the complex plane and $z$ a complex vector with $n$ components. The product for $n$-dimensional vectors $x$, $x'$ is defined as $x \cdot x' \equiv x^T Q x'$, where $Q$ is an $n \times n$-dimensional integral symmetric matrix. We assume that the quadratic form $x \cdot x'$ is non-degenerate and denote the determinant of $Q$ by $\Delta \equiv \det Q$. Special vectors are then defined as integral vectors $x$ with $n$ components such that $Qx$ is divisible by $\Delta$, i.e. each of its components is divisible by $\Delta$. In (A.1) the vectors $g$, $h$ and $\alpha$ are special vectors. Lastly, $N$ is a positive integer and the sum is over the vectors $m$ that are congruent to $\alpha$ modulo $N\Delta$, i.e. each component of $m - \alpha$ is divisible by $N\Delta$. For notational convenience we use this unconventional notation.

The theta functions we consider here are generalizations of the theta functions considered in [38] for the case of matrices $Q$ that are not necessarily positive-definite. Here we derive properties for these more general theta functions adapting what was done in [38]. In particular this will result in the modular transformation of the functions (A.1), i.e.

$$\mathcal{O}(\gamma)\Theta_{g,h}(\tau, \overline{\tau}, z; \alpha, N) \equiv \Theta_{g,h}\left(\frac{a\tau + b}{c\tau + d}, \frac{a\overline{\tau} + b}{c\overline{\tau} + d}, \frac{z_-}{c\tau + d} + \frac{z_+}{c\overline{\tau} + d}; \alpha, N\right), \tag{A.2}$$

where

$$\gamma = \begin{pmatrix} a & b \\ c & d \end{pmatrix} \tag{A.3}$$

is an integral matrix with determinant 1.

Let us now derive these properties. First of all, for special vectors $l$

$$\Theta_{g+2l,h}(\tau,\overline{\tau},z;\alpha,N) = \Theta_{g,h}(\tau,\overline{\tau},z;\alpha+l,N)\,, \tag{A.4}$$

$$\Theta_{g,h+2l}(\tau,\overline{\tau},z;\alpha,N) = \Theta_{g,h}(\tau,\overline{\tau},z;\alpha,N)\,. \tag{A.5}$$

For an arbitrary integral vector $k$ we find that

$$\Theta_{g,h}(\tau,\overline{\tau},z;\alpha+N\Delta k,N) = (-1)^{\frac{1}{\Delta}h\cdot k}\Theta_{g,h}(\tau,\overline{\tau},z;\alpha,N)\,. \tag{A.6}$$

Let's now consider some special modular transformations (A.2). First of all the transformation $\mathcal{O}(\gamma)$ with

$$\gamma = \begin{pmatrix} -1 & 0 \\ 0 & -1 \end{pmatrix}, \tag{A.7}$$

which corresponds to replacing $z$ by $-z$, is given by

$$
\begin{aligned}
\Theta_{g,h}(\tau,\overline{\tau},-z;\alpha,N) &= \sum_{m\equiv\alpha\,(N\Delta)} (-1)^{\frac{1}{N\Delta^2}h\cdot(m-\alpha)}e^{-\pi i\tau\frac{1}{N\Delta^2}(m+\frac{1}{2}g)^2_-} \\
&\quad \times e^{-\pi i\overline{\tau}\frac{1}{N\Delta^2}(m+\frac{1}{2}g)^2_+}e^{\frac{2\pi i}{\Delta}(-m-\frac{1}{2}g)\cdot z} \\
&= \sum_{m'\equiv-\alpha-g\,(N\Delta)} (-1)^{\frac{1}{N\Delta^2}h\cdot(m'+g+\alpha)}e^{-\pi i\tau\frac{1}{N\Delta^2}(m'+\frac{1}{2}g)^2_-} \\
&\quad \times e^{-\pi i\overline{\tau}\frac{1}{N\Delta^2}(m'+\frac{1}{2}g)^2_+}e^{\frac{2\pi i}{\Delta}(m'+\frac{1}{2}g)\cdot z} \\
&= \Theta_{g,h}(\tau,\overline{\tau},z;-\alpha-g,N)\,, \tag{A.8}
\end{aligned}
$$

where we relabeled $m' = -m - g$ for the second equality sign. Another transformation we would like to work out is given by $\mathcal{O}(\gamma)$ with

$$\gamma = \begin{pmatrix} 1 & b \\ 0 & 1 \end{pmatrix}, \tag{A.9}$$

which is one of the generators of the modular group. Using that $m = \alpha+N\Delta k$ for $k \in \mathbb{Z}^n$ it follows that

$$\Theta_{g,h}(\tau+b,\overline{\tau}+b,z;\alpha,N) = e^{-\pi i\frac{b}{N\Delta^2}(\alpha+\frac{1}{2}g)^2}\sum_{k\in\mathbb{Z}^n}(-1)^{\frac{1}{N\Delta^2}h\cdot N\Delta k}(-1)^{-\frac{b}{N\Delta^2}(N^2\Delta^2 k^2+N\Delta k\cdot g)}$$

$$\times e^{-\pi i\tau\frac{1}{N\Delta^2}(\alpha+N\Delta k+\frac{1}{2}g)^2_-}e^{-\pi i\overline{\tau}\frac{1}{N\Delta^2}(\alpha+N\Delta k+\frac{1}{2}g)^2_+}e^{\frac{2\pi i}{\Delta}(\alpha+N\Delta k+\frac{1}{2}g)\cdot z}\,. \tag{A.10}$$

We now define $t$, which is the vector whose components are the diagonal elements of $Q$, and $v = \Delta Q^{-1}t$ which is a special vector. For an arbitrary integral vector $k$ we find that

$$\Delta^{-1}v\cdot k = k^T t \equiv k^T Q k = k^2 \ (\mathrm{mod}\,2)\,, \tag{A.11}$$

which we can use to rewrite

$$(-1)^{\frac{1}{N\Delta^2}h\cdot N\Delta k}(-1)^{-\frac{b}{N\Delta^2}(N^2\Delta^2 k^2+N\Delta k\cdot g)} = (-1)^{\frac{1}{N\Delta^2}(h-bNv-bg)\cdot N\Delta k}\,. \tag{A.12}$$

Using this in (A.10) and rewriting the sum in modulus form we find that

$$
\Theta_{g,h}(\tau+b,\overline{\tau}+b,z;\alpha,N) = e^{-\pi i \frac{b}{N\Delta^2}(\alpha+\frac{1}{2}g)^2} \sum_{m\equiv\alpha\,(N\Delta)} (-1)^{\frac{1}{N\Delta^2}(h-bg-bNv)\cdot(m-\alpha)}
$$

$$
\times\, e^{-\pi i \tau \frac{1}{N\Delta^2}(m+\frac{1}{2}g)^2_-} e^{-\pi i \overline{\tau} \frac{1}{N\Delta^2}(m+\frac{1}{2}g)^2_+} e^{\frac{2\pi i}{\Delta}(m+\frac{1}{2}g)\cdot z}
$$

$$
= e^{-\pi i \frac{b}{N\Delta^2}(\alpha+\frac{1}{2}g)^2} \Theta_{g,h-bg-bNv}(\tau,\overline{\tau},z;\alpha,N). \qquad (\mathrm{A}.13)
$$

The last transformation we work out before considering the most general modular transformation is given by $\mathcal{O}(\gamma)$ with

$$
\gamma = \begin{pmatrix} 0 & -1 \\ 1 & 0 \end{pmatrix}, \qquad (\mathrm{A}.14)
$$

which is the generator $S$ of the modular group. In order to work out the transformation of the theta function under $\mathcal{O}(\gamma)$ we make use of Poisson's summation formula. Namely for a continuous function $f(k) \equiv f(k_1, ..., k_n)$ on $\mathbb{R}^n$ that has continuous first and second order partial derivatives and has the property that $\sum_{k\in\mathbb{Z}^n} f(k+u)$, $\sum_{k\in\mathbb{Z}^n} \partial_l f(k+u)$, as well as $\sum_{k\in\mathbb{Z}^n} \partial_l \partial_m f(k+u)$ all converge absolutely and uniformly for $0 \leq u_i \leq 1$, one has that

$$
\sum_{k\in\mathbb{Z}^n} f(k) = \sum_{k\in\mathbb{Z}^n} \int_{-\infty}^{\infty} du_1 \cdots \int_{-\infty}^{\infty} du_n\, e^{-2\pi i k^T u} f(u). \qquad (\mathrm{A}.15)
$$

Writing $m = \alpha + N\Delta k$ we find

$$
\Theta_{g,h}\left(-\frac{1}{\tau},-\frac{1}{\overline{\tau}},\frac{z_-}{\tau}+\frac{z_+}{\overline{\tau}};\alpha,N\right) = \sum_{k\in\mathbb{Z}^n} f(k), \qquad (\mathrm{A}.16)
$$

where

$$
f(k) = (-1)^{-\frac{1}{\Delta}h\cdot k} e^{\pi i \frac{1}{\tau}\frac{1}{N\Delta^2}(\alpha+N\Delta k+\frac{1}{2}g)^2_-} e^{\pi i \frac{1}{\overline{\tau}}\frac{1}{N\Delta^2}(\alpha+N\Delta k+\frac{1}{2}g)^2_+} e^{\frac{2\pi i}{\Delta}(\alpha+N\Delta k+\frac{1}{2}g)\cdot\left(\frac{z_-}{\tau}+\frac{z_+}{\overline{\tau}}\right)}.
$$

$$
(\mathrm{A}.17)
$$

Applying (A.15), using

$$
e^{-2\pi i k^T u} f(u) = e^{-2\pi i (k^T+\frac{1}{2}\frac{1}{\Delta}h\cdot)u} e^{\pi i \frac{1}{\tau}\frac{1}{N\Delta^2}(\alpha+N\Delta u+\frac{1}{2}g)^2_-} e^{\pi i \frac{1}{\overline{\tau}}\frac{1}{N\Delta^2}(\alpha+N\Delta u+\frac{1}{2}g)^2_+} e^{\frac{2\pi i}{\Delta}(\alpha+N\Delta u+\frac{1}{2}g)\cdot\left(\frac{z_-}{\tau}+\frac{z_+}{\overline{\tau}}\right)},
$$

$$
(\mathrm{A}.18)
$$

and replacing $\alpha + N\Delta u + \frac{1}{2}g$ by $N\Delta u$, we find that

$$
\Theta_{g,h}\left(-\frac{1}{\tau},-\frac{1}{\overline{\tau}},\frac{z_-}{\tau}+\frac{z_+}{\overline{\tau}};\alpha,N\right) = \sum_{k\in\mathbb{Z}^n} e^{2\pi i \frac{1}{N\Delta}(k^T+\frac{1}{2}\frac{1}{\Delta}h\cdot)(\alpha+\frac{1}{2}g)} \int_{-\infty}^{\infty} du_1 \cdots \int_{-\infty}^{\infty} du_n
$$

$$
\times\, e^{-2\pi i (k^T+\frac{1}{2}\frac{1}{\Delta}h\cdot)u} e^{\pi i \frac{1}{\tau}N u^2_-} e^{\pi i \frac{1}{\overline{\tau}}N u^2_+} e^{2\pi i N u\cdot\left(\frac{z_-}{\tau}+\frac{z_+}{\overline{\tau}}\right)}. \qquad (\mathrm{A}.19)
$$

Defining $y = \Delta Q^{-1}k$, we can complete the square of the exponent:

$$
u^2_- + 2u_-\cdot z_- - 2\tau\frac{1}{N\Delta}u_-\cdot\left(y+\tfrac{1}{2}h\right)_- = \left[u+z-\frac{\tau}{N\Delta}\left(y+\tfrac{1}{2}h\right)\right]^2_- - \left[z-\frac{\tau}{N\Delta}\left(y+\tfrac{1}{2}h\right)\right]^2_-,
$$

$$
u^2_+ + 2u_+\cdot z_+ - 2\overline{\tau}\frac{1}{N\Delta}u_+\cdot\left(y+\tfrac{1}{2}h\right)_+ = \left[u+z-\frac{\overline{\tau}}{N\Delta}\left(y+\tfrac{1}{2}h\right)\right]^2_+ - \left[z-\frac{\overline{\tau}}{N\Delta}\left(y+\tfrac{1}{2}h\right)\right]^2_+.
$$

$$
(\mathrm{A}.20)
$$

The integrals then result in

$$
\int_{-\infty}^{\infty} du_1 \cdots \int_{-\infty}^{\infty} du_n \, e^{\pi i \frac{1}{\tau} N\left[u+z+\frac{\tau}{N\Delta}\left(y+\frac{1}{2}h\right)\right]^2_-} e^{\pi i \frac{1}{\bar{\tau}} N\left[u+z+\frac{\bar{\tau}}{N\Delta}\left(y+\frac{1}{2}h\right)\right]^2_+}
$$

$$
= \frac{1}{\sqrt{\det(-Q_-)}} \left(\frac{-i\tau}{N}\right)^{b_-/2} \frac{1}{\sqrt{\det Q_+}} \left(\frac{i\bar{\tau}}{N}\right)^{b_+/2}
$$

$$
= \frac{1}{\sqrt{|\Delta|}} \left(\frac{-i\tau}{N}\right)^{b_-/2} \left(\frac{i\bar{\tau}}{N}\right)^{b_+/2} , \tag{A.21}
$$

where $b_-$ ($b_+$) is the number of negative (positive) eigenvalues of $Q$ and $\det(-Q_-)$ ($\det Q_+$) is the absolute value of the product of the negative (positive) eigenvalues of $Q$. Hence from (A.19) we find that

$$
\Theta_{g,h}\left(-\frac{1}{\tau}, -\frac{1}{\bar{\tau}}, \frac{z_-}{\tau} + \frac{z_+}{\bar{\tau}}; \alpha, N\right) = \frac{1}{\sqrt{|\Delta|}} \left(\frac{-i\tau}{N}\right)^{b_-/2} \left(\frac{i\bar{\tau}}{N}\right)^{b_+/2} \sum_{y}^{*} e^{2\pi i \frac{1}{N\Delta^2}\left(\alpha+\frac{1}{2}g\right)\cdot\left(y+\frac{1}{2}h\right)}
$$

$$
\times e^{-\pi i \frac{1}{\tau} N\left[z-\frac{\tau}{N\Delta}\left(y+\frac{1}{2}h\right)\right]^2_-} e^{-\pi i \frac{1}{\bar{\tau}} N\left[z-\frac{\bar{\tau}}{N\Delta}\left(y+\frac{1}{2}h\right)\right]^2_+} , \tag{A.22}
$$

where the $*$ indicates that summation is over special vectors. Note that all these vectors can be uniquely written as $y = \beta + N\Delta k$, where $\beta$ is a special vector in the set of vectors that are incongruent modulo $N\Delta$ and $k \in \mathbb{Z}^n$. We will use this to rewrite the sum in (A.22) and obtain an expression in terms of theta functions. To this end we also need the equality

$$
e^{2\pi i \frac{1}{N\Delta^2}\left(\alpha+\frac{1}{2}g\right)\cdot\left(y+\frac{1}{2}h\right)} = e^{2\pi i \frac{1}{N\Delta^2}\left[\left(\beta+\frac{1}{2}h\right)\cdot\left(\alpha+\frac{1}{2}g\right)+N\Delta k\cdot\alpha+\frac{1}{2}N\Delta k\cdot g\right]}
$$

$$
= e^{2\pi i \frac{1}{N\Delta^2}\left(\beta+\frac{1}{2}h\right)\cdot\left(\alpha+\frac{1}{2}g\right)}(-1)^{\frac{1}{N\Delta^2} g\cdot(y-\beta)} . \tag{A.23}
$$

Using this identity we can rewrite (A.22):

$$
\Theta_{g,h}\left(-\frac{1}{\tau}, -\frac{1}{\bar{\tau}}, \frac{z_-}{\tau} + \frac{z_+}{\bar{\tau}}; \alpha, N\right)
$$

$$
= \frac{1}{\sqrt{|\Delta|}} \left(\frac{-i\tau}{N}\right)^{b_-/2} \left(\frac{i\bar{\tau}}{N}\right)^{b_+/2} e^{-\pi i \frac{1}{\tau} N z_-^2} e^{-\pi i \frac{1}{\bar{\tau}} N z_+^2} \sum_{\beta\,(N\Delta)}^{*} e^{2\pi i \frac{1}{N\Delta^2}\left(\beta+\frac{1}{2}h\right)\cdot\left(\alpha+\frac{1}{2}g\right)}
$$

$$
\times \sum_{y\equiv\beta\,(N\Delta)} (-1)^{\frac{1}{N\Delta^2} g\cdot(y-\beta)} e^{-\pi i \tau \frac{1}{N\Delta^2}\left(y+\frac{1}{2}h\right)^2_-} e^{-\pi i \bar{\tau} \frac{1}{N\Delta^2}\left(y+\frac{1}{2}h\right)^2_+} e^{\frac{2\pi i}{\Delta} z\cdot\left(y+\frac{1}{2}h\right)}
$$

$$
= \frac{1}{\sqrt{|\Delta|}} \left(\frac{-i\tau}{N}\right)^{b_-/2} \left(\frac{i\bar{\tau}}{N}\right)^{b_+/2} e^{-\pi i \frac{1}{\tau} N z_-^2} e^{-\pi i \frac{1}{\bar{\tau}} N z_+^2}
$$

$$
\times \sum_{\beta\,(N\Delta)}^{*} e^{2\pi i \frac{1}{N\Delta^2}\left(\beta+\frac{1}{2}h\right)\cdot\left(\alpha+\frac{1}{2}g\right)} \Theta_{h,g}(\tau, \bar{\tau}, z; \beta, N) , \tag{A.24}
$$

where we now sum over special vectors $\beta$ that are incongruent modulo $N\Delta$.

We now prove one last identity before figuring out the transformation under an arbitrary element of the modular group. Let $M$ be a positive integer, then

$$\sum_{\substack{m\,(MN\Delta)\\ m\equiv\alpha\,(N\Delta)}} (-1)^{\frac{1}{N\Delta^2}h\cdot(m-\alpha)}\Theta_{g,Mh}(M\tau, M\bar\tau, z; m, MN) \tag{A.25}$$

$$= \sum_{\substack{m\,(MN\Delta)\\ m\equiv\alpha\,(N\Delta)}}\sum_{n\equiv m\,(MN\Delta)} (-1)^{\frac{1}{N\Delta^2}h\cdot(n-\alpha)} e^{-\pi i\tau\frac{1}{N\Delta^2}(n+\frac{1}{2}g)^2_-} e^{-\pi i\bar\tau\frac{1}{N\Delta^2}(n+\frac{1}{2}g)^2_+} e^{\frac{2\pi i}{\Delta}(n+\frac{1}{2}g)\cdot z}\,.$$

Notice that the summation can be replaced by a sum over $n\equiv\alpha\,(\mathrm{mod}\,N\Delta)$ such that

$$\sum_{\substack{m\,(MN\Delta)\\ m\equiv\alpha\,(N\Delta)}} (-1)^{\frac{1}{N\Delta^2}h\cdot(m-\alpha)}\Theta_{g,Mh}(M\tau, M\bar\tau, z; m, MN) = \Theta_{g,h}(\tau,\bar\tau, z;\alpha, N)\,. \tag{A.26}$$

Let's now consider the modular transformation $\mathcal{O}(\gamma)$ corresponding to (A.3) for $c\neq 0$. Then from (A.26) with $M=|c|$ we find that

$$\mathcal{O}(\gamma)\Theta_{g,h}(\tau,\bar\tau, z;\alpha, N) = \sum_{\substack{m\,(cN\Delta)\\ m\equiv\alpha\,(N\Delta)}} (-1)^{\frac{1}{N\Delta^2}h\cdot(m-\alpha)}$$

$$\times\,\Theta_{g,|c|h}\left(a\,\mathrm{sgn}\,c - \frac{\mathrm{sgn}\,c}{c\tau+d}, a\,\mathrm{sgn}\,c - \frac{\mathrm{sgn}\,c}{c\bar\tau+d}, \frac{z_-}{c\tau+d} + \frac{z_+}{c\bar\tau+d}; m, |c|N\right)\,, \tag{A.27}$$

where we also used that

$$\frac{a}{c} - \frac{1}{c(c\tau+d)} = \frac{a\tau+b}{c\tau+d}\,, \qquad \frac{a}{c} - \frac{1}{c(c\bar\tau+d)} = \frac{a\bar\tau+b}{c\bar\tau+d}\,. \tag{A.28}$$

With (A.13) we can rewrite (A.27) as

$$\mathcal{O}(\gamma)\Theta_{g,h}(\tau,\bar\tau, z;\alpha, N) = \sum_{\substack{m\,(cN\Delta)\\ m\equiv\alpha\,(N\Delta)}} (-1)^{\frac{1}{N\Delta^2}h\cdot(m-\alpha)} e^{-\pi i\frac{a}{cN\Delta^2}(m+\frac{1}{2}g)^2} \tag{A.29}$$

$$\times\,\Theta_{g,g_1}\left(-\frac{\mathrm{sgn}\,c}{c\tau+d}, -\frac{\mathrm{sgn}\,c}{c\bar\tau+d}, \frac{z_-}{c\tau+d} + \frac{z_+}{c\bar\tau+d}; m, |c|N\right)\,,$$

where using (A.5) $g_1$ can be written as $g_1 = ch + ag + acNv$. According to (A.24) we have that

$$\Theta_{g,g_1}\left(-\frac{\mathrm{sgn}\,c}{c\tau+d}, -\frac{\mathrm{sgn}\,c}{c\bar\tau+d}, \frac{z_-}{c\tau+d} + \frac{z_+}{c\bar\tau+d}; m, |c|N\right) = W\sum_{\beta\,(cN\Delta)}^{*} e^{2\pi i\frac{1}{|c|N\Delta^2}(\beta+\frac{1}{2}g_1)\cdot(m+\frac{1}{2}g)}$$

$$\times\,\Theta_{g_1,g}\left((c\tau+d)\mathrm{sgn}\,c, (c\bar\tau+d)\mathrm{sgn}\,c, z\,\mathrm{sgn}\,c; \beta, |c|N\right)\,, \tag{A.30}$$

where

$$W = \frac{1}{\sqrt{|\Delta|}}\left(\frac{-i(c\tau+d)\mathrm{sgn}\,c}{|c|N}\right)^{b_-/2}\left(\frac{i\,(c\bar\tau+d)\,\mathrm{sgn}\,c}{|c|N}\right)^{b_+/2} e^{-\pi i\frac{cNz_-^2}{c\tau+d}} e^{-\pi i\frac{cNz_+^2}{c\bar\tau+d}}\,. \tag{A.31}$$

We now again use (A.13) to write

$$\Theta_{g_1,g}\big((c\tau+d)\mathrm{sgn}\,c,(c\bar\tau+d)\mathrm{sgn}\,c,z\,\mathrm{sgn}\,c;\beta,|c|N\big)=e^{-\pi i\frac{d}{cN\Delta^2}(\beta+\frac{1}{2}g_1)^2}\Theta_{g_1,h_0}(|c|\tau,|c|\bar\tau,z\,\mathrm{sgn}\,c;\beta,|c|N)\,,$$
$$(A.32)$$

where, using (A.5), $h_0$ can be written as

$$h_0=(1+da)g+dch+dc(a+1)Nv\,. \tag{A.33}$$

The numbers $1+ad\equiv bc\ (\mathrm{mod}\,2)$ and $cd(a+1)\equiv bcd\ (\mathrm{mod}\,2)$, where we used that $ad-bc=1$ and thus $d\equiv ad-bcd\ (\mathrm{mod}\,2)$. Again applying (A.5), we then find that $h_0=ch_1$ where

$$h_1=bg+dh+bdNv\,. \tag{A.34}$$

Now we specify to $c>0$. Substitution of (A.32) and (A.30) in (A.29) results in

$$\mathcal{O}(\gamma)\Theta_{g,h}(\tau,\bar\tau,z;\alpha,N)=W\sum_{\beta\,(cN\Delta)}^{*}\varphi_{g_1,g,h}(\alpha,\beta;N,\gamma)\Theta_{g_1,ch_1}(c\tau,c\bar\tau,z;\beta,cN)\,, \tag{A.35}$$

where we defined

$$\varphi_{g_1,g,h}(\alpha,\beta;N,\gamma)\equiv\sum_{\substack{m\,(cN\Delta)\\ m\equiv\alpha\,(N\Delta)}}(-1)^{\frac{1}{N\Delta^2}h\cdot(m-\alpha)}e^{-\pi i\left[\frac{a}{cN\Delta^2}(m+\frac{1}{2}g)^2-\frac{2}{cN\Delta^2}(\beta+\frac{1}{2}g_1)\cdot(m+\frac{1}{2}g)+\frac{d}{cN\Delta^2}(\beta+\frac{1}{2}g_1)^2\right]}\,.$$
$$(A.36)$$

Let's now consider the case $c<0$. Using (A.8) we find that

$$\Theta_{g_1,h_0}(|c|\tau,|c|\bar\tau,-z;\beta,|c|N)=\Theta_{g_1,h_0}(|c|\tau,|c|\bar\tau,z;-\beta-g_1,|c|N) \tag{A.37}$$

which when substituted with (A.32) and (A.30) in (A.29) yields

$$\mathcal{O}(\gamma)\Theta_{g,h}(\tau,\bar\tau,z;\alpha,N)=\sum_{\substack{m\,(cN\Delta)\\ m\equiv\alpha\,(N\Delta)}}(-1)^{\frac{1}{N\Delta^2}h\cdot(m-\alpha)}e^{-\pi i\frac{a}{cN\Delta^2}(m+\frac{1}{2}g)^2}W \tag{A.38}$$

$$\times\sum_{\beta\,(cN\Delta)}^{*}e^{-2\pi i\frac{1}{cN\Delta^2}(\beta+\frac{1}{2}g_1)\cdot(m+\frac{1}{2}g)}e^{-\pi i\frac{d}{cN\Delta^2}(\beta+\frac{1}{2}g_1)^2}\Theta_{g_1,h_0}(|c|\tau,|c|\bar\tau,z;-\beta-g_1,|c|N)\,.$$

Changing the summation variable $\beta$ to $\beta'=-\beta-g_1$ does not change the range of summation, so we can write

$$\mathcal{O}(\gamma)\Theta_{g,h}(\tau,\bar\tau,z;\alpha,N)\;=\;W\sum_{\substack{\beta'\,(cN\Delta)}}^{*}\sum_{\substack{m\,(cN\Delta)\\ m\equiv\alpha\,(N\Delta)}}(-1)^{\frac{1}{N\Delta^2}h\cdot(m-\alpha)}e^{-\pi i\frac{a}{cN\Delta^2}(m+\frac{1}{2}g)^2}$$

$$\times\,e^{2\pi i\frac{1}{cN\Delta^2}(\beta'+\frac{1}{2}g_1)\cdot(m+\frac{1}{2}g)}e^{-\pi i\frac{d}{cN\Delta^2}(\beta'+\frac{1}{2}g_1)^2}\Theta_{g_1,h_0}(|c|\tau,|c|\bar\tau,z;\beta',|c|N)$$

$$=\;W\sum_{\beta'\,(cN\Delta)}^{*}\varphi_{g_1,g,h}(\alpha,\beta';N,\gamma)\Theta_{g_1,h_0}(|c|\tau,|c|\bar\tau,z;\beta',|c|N)\,. \tag{A.39}$$

This is the same expression as we had for $c > 0$. For the sum $\varphi$ the following identity holds [38][17]:

$$\varphi_{g_1,g,h}(\alpha, \beta + N\Delta k; N, \gamma) = (-1)^{\frac{1}{\Delta}(bg+dh+bdNv)\cdot k}\varphi_{g_1,g,h}(\alpha, \beta; N, \gamma), \tag{A.40}$$

where $k$ is an arbitrary integral vector. Hence if $\beta' \equiv \beta \pmod{\Delta}$, we find that

$$\varphi_{g_1,g,h}(\alpha, \beta'; N, \gamma) = (-1)^{\frac{1}{N\Delta^2}h_1\cdot(\beta-\beta')}\varphi_{g_1,g,h}(\alpha, \beta; N, \gamma) \tag{A.41}$$

such that (A.39), after reparameterizing the summation variables, becomes

$$
\begin{aligned}
\mathcal{O}(\gamma)\Theta_{g,h}(\tau, \bar{\tau}, z; \alpha, N) &= W \sum_{\beta\,(N\Delta)}^{*} \sum_{\substack{\beta'\,(cN\Delta) \\ \beta' \equiv \beta\,(N\Delta)}} \varphi_{g_1,g,h}(\alpha, \beta'; N, \gamma)\Theta_{g_1,h_0}(|c|\tau, |c|\bar{\tau}, z; \beta', |c|N) \\[2mm]
&= W \sum_{\beta\,(N\Delta)}^{*} \varphi_{g_1,g,h}(\alpha, \beta; N, \gamma) \\[2mm]
&\quad \times \sum_{\substack{\beta'\,(cN\Delta) \\ \beta' \equiv \beta\,(N\Delta)}} (-1)^{\frac{1}{N\Delta^2}h_1\cdot(\beta'-\beta)}\Theta_{g_1,|c|h_1}(|c|\tau, |c|\bar{\tau}, z; \beta', |c|N).
\end{aligned}
\tag{A.42}
$$

We then apply (A.26) to get

$$\mathcal{O}(\gamma)\Theta_{g,h}(\tau, \bar{\tau}, z; \alpha, N) = W \sum_{\beta\,(N\Delta)}^{*} \varphi_{g_1,g,h}(\alpha, \beta; N, \gamma)\Theta_{g_1,h_1}(\tau, \bar{\tau}, z; \beta, N). \tag{A.43}$$

The only modular transformations that we have not considered yet have $c = 0$. Note that this implies $a = d = 1$ or $a = d = -1$. The first case gives with (A.13)

$$\mathcal{O}(\gamma)\Theta_{g,h}(\tau, \bar{\tau}, z; \alpha, N) = e^{-\pi i \frac{b}{N\Delta^2}(\alpha+\frac{1}{2}g)^2}\Theta_{g_1,h_1}(\tau, \bar{\tau}, z; \alpha, N). \tag{A.44}$$

When $a = d = -1$ we find with (A.13) that

$$\mathcal{O}(\gamma)\Theta_{g,h}(\tau, \bar{\tau}, z; \alpha, N) = e^{\pi i \frac{b}{N\Delta^2}(\alpha+\frac{1}{2}g)^2}\Theta_{g,h-bg-bNv}(\tau, \bar{\tau}, -z; \alpha, N). \tag{A.45}$$

Application of (A.8), (A.4) and (A.5) yields:

$$\mathcal{O}(\gamma)\Theta_{g,h}(\tau, \bar{\tau}, z; \alpha, N) = e^{\pi i \frac{b}{N\Delta^2}(\alpha+\frac{1}{2}g)^2}\Theta_{g_1,h_1}(\tau, \bar{\tau}, z; -\alpha, N). \tag{A.46}$$

---

[17]In [38] one has to consider $\varphi$ corresponding to the modular transformation with $-\gamma$, $-\beta$ to recover the function $\varphi$ here. The proof of the property (A.40) there is also valid for non-degenerate matrices $Q$ that are not positive-definite.

# B    Modular transformation theta functions elliptic genus

In this appendix we derive how the theta functions (2.25)

$$\Theta_\mu(\tau, \overline{\tau}, z) = \sum_{k \in \mathbb{Z}^n} (-1)^{p \cdot (\mu + k + \frac{1}{2}p)} e^{-\pi i \tau (\mu + k + \frac{1}{2}p)^2_-} e^{-\pi i \overline{\tau}(\mu + k + \frac{1}{2}p)^2_+} e^{2\pi i (\mu + k + \frac{1}{2}p) \cdot z} \tag{B.1}$$

that we find in the expansion of the elliptic genus transform under modular transformations. Note that in (B.1) we have chosen a basis for $\Lambda$. We can derive the modular properties by writing (B.1) in the form of the general theta functions (A.1) of the previous appendix:

$$\Theta_\mu(\tau, \overline{\tau}, z) = (-1)^{\frac{1}{2}p^2 + p \cdot \mu} \Theta_{\Delta(p+2\mu), \Delta p}(\tau, \overline{\tau}, z; 0, 1) . \tag{B.2}$$

Here we identified the matrix with components $d_{ab}$ as $Q$. Furthermore, since $p \in \Lambda$, we find that $Qh = \Delta Qp$ is divisible by $\Delta$ and thus that $h$ is a special vector. To show that $g = \Delta(p + 2\mu)$ is a special vector requires slightly more work since $\mu \in \Lambda^*/\Lambda$. Elements of $\Lambda$ are represented by vectors $k \in \mathbb{Z}^n$. The dual lattice $\Lambda^*$ is defined by all vectors $l$ such that $l \cdot k = \in \mathbb{Z}$ for all $k \in \mathbb{Z}^n$. We observe that this is equivalent to $Ql \in \mathbb{Z}^n$, which in term is equal to $l = Q^{-1}w$ for vectors $w \in \mathbb{Z}^n$. We can write $Q^{-1} = \frac{1}{\Delta}Q'$ where $Q'$ is a matrix over $\mathbb{Z}$. We thus find that $l = \frac{w'}{\Delta}$, where $w' = Q'w \in \mathbb{Z}^n$. Now $Qw' = QQ'w = \Delta w$, which immediately implies that $w'$ must be a special vector. We thus found that all vectors in $\Lambda^*$ are of the form $\frac{w'}{\Delta}$ for special vectors $w'$. We now determine $\Lambda^*/\Lambda$ by writing $w' = \Delta k + k_1$, where $k \in \mathbb{Z}^n$ and $k_1$ takes values $(\mathrm{mod}\, \Delta)$. Then

$$\frac{w'}{\Delta} = k + \frac{k_1}{\Delta} \equiv \frac{k_1}{\Delta} \,(\mathrm{mod}\, \Lambda) . \tag{B.3}$$

Hence vectors of $\Lambda^*/\Lambda$ are of the form $k_1/\Delta$ for special vectors $k_1 \,(\mathrm{mod}\, \Delta)$. Then $\Delta\mu$ is a special vector which implies that $g$ is a special vector.

We now use the previous appendix to derive what happens under a general modular transformation with $c \neq 0$. Using the identification (B.2) and the modular transformation of the general theta functions (A.43) we find that

$$\mathcal{O}(\gamma)\Theta_\mu(\tau, \overline{\tau}, z) = (-1)^{\frac{1}{2}p^2 + p \cdot \mu} W \sum_{\beta\,(\Delta)}^{*} \varphi_{g_1, \Delta(p+2\mu), \Delta p}(0, \beta; 1, \gamma) \Theta_{g_1, h_1}(\tau, \overline{\tau}, z; \beta, 1) , \tag{B.4}$$

where $W$ is given by (A.31) for $N = 1$ and the vectors $g_1$, $h_1$ are equal to

$$g_1 = c\Delta p + a\Delta(p + 2\mu) + acv , \qquad h_1 = d\Delta p + b\Delta(p + 2\mu) + bdv . \tag{B.5}$$

The theta function at the right-hand side of (B.4) now needs to be rewritten in terms of the theta functions (B.1) using the identification (B.2). With (A.4) we rewrite

$$\Theta_{g_1, h_1}(\tau, \overline{\tau}, z; \beta, 1) = \Theta_{g_1 + 2\beta, h_1}(\tau, \overline{\tau}, z; 0, 1) . \tag{B.6}$$

We now want to show that $h_1 = \Delta p + 2l$, where $l$ is a special vector. This way we can apply (A.5). Since $ad - bc = 1$ there are two cases to consider: 1) both $d$ and $b$ are odd, 2) one of them is even and the other one is odd. In the latter case it is clear that $h_1$ is of the required form because $d + b - 1$, $bd$ are even and $\Delta p$, $\Delta \mu$, $v$ are special vectors. In the first case we can write $h_1 = \Delta p + 2l_1 + 2l_2$, where

$$l_1 = \tfrac{1}{2}\left[(d + b - 2)\Delta p + 2b\Delta\mu + (bd - 1)v\right] \tag{B.7}$$

is a special vector and $l_2 = \tfrac{1}{2}(\Delta p + v)$. Now for an arbitrary integral vector $k$ we have that

$$\Delta k \cdot p + k \cdot v \equiv \Delta k \cdot p + \Delta k^2 \equiv 0 \;(\mathrm{mod}\,2)\,, \tag{B.8}$$

where we have used (A.11) and (2.12). Note that this implies that $l_2$ is an integral vector. To prove that it is also a special vector, we need to show that $Ql_2 = \tfrac{1}{2}\Delta(Qp + t) \equiv 0 \;(\mathrm{mod}\,\Delta)$. For an arbitrary integral vector $k$

$$k \cdot Qp + k \cdot t = k' \cdot p + \frac{1}{\Delta}k' \cdot v \equiv k' \cdot p + k'^2 \equiv 0 \;(\mathrm{mod}\,2) \tag{B.9}$$

where $k' = Qk \in \mathbb{Z}^n$. This proves that $Qp + t$ is divisible by 2 such that $Ql_2$ is divisible by $\Delta$. Thus $l_2$ is a special vector. This means that in both cases we can apply (A.5) and write

$$\Theta_{g_1,h_1}(\tau,\bar{\tau},z;\beta,1) = \Theta_{g_1+2\beta,\Delta p}(\tau,\bar{\tau},z;0,1)\,. \tag{B.10}$$

The last thing we need to do is to show that we can write

$$g_1 + 2\beta = \Delta p + 2l + 2\beta \tag{B.11}$$

for a special vector $l$. Then $\beta' = l + \beta$ is special, which can be used to rewrite the sum in (B.4) as a sum over special vectors $\beta' \;(\mathrm{mod}\,\Delta)$. From (B.5) it follows that

$$l = \tfrac{1}{2}\left[(c + a - 1)\Delta p + 2a\Delta\mu + acv\right]\,. \tag{B.12}$$

We again consider two different cases: 1) both $c$ and $a$ are odd, 2) one of them is even and the other one is odd. In the second case it is clear that $l$ is a special vector because $c + a - 1$, $ac$ are even and $\Delta p$, $\Delta \mu$ and $v$ are special vectors. In the first case we observe that $l = l_1 + l_2$ where

$$l_1 = \tfrac{1}{2}\left[(c + a - 2)\Delta p + 2a\Delta\mu + (ac - 1)v\right] \tag{B.13}$$

is a special vector and $l_2 = \tfrac{1}{2}(\Delta p + v)$. Since we have already shown that $l_2$ is a special vector we find that we always have the splitting (B.11). With all of the above we can rewrite (B.4) as

$$\mathcal{O}(\gamma)\Theta_\mu(\tau,\bar{\tau},z) = (-1)^{\frac{1}{2}p^2+p\cdot\mu}\,W\,\sum_{\beta'-l(\Delta)}^{*}\varphi_{g_1,\Delta(p+2\mu),\Delta p}(0,\beta'-l;1,\gamma)\Theta_{\Delta p+2\beta',\Delta p}(\tau,\bar{\tau},z;0,1)\,. \tag{B.14}$$

Since $l$ is a special vector, the range of values for $\beta'$ does not change. Also from the definition of $\varphi$ (A.36) it straightforwardly follows that

$$\varphi_{g_1,g,h}(0, \beta' - l; 1, \gamma) = \varphi_{g_1-2l,g,h}(0, \beta'; 1, \gamma) = \varphi_{\Delta p, \Delta(p+2\mu), \Delta p}(0, \beta'; 1, \gamma) \, . \tag{B.15}$$

Hence, (B.14) becomes

$$\mathcal{O}(\gamma)\Theta_\mu(\tau, \overline{\tau}, z) = (-1)^{\frac{1}{2}p^2 + p \cdot \mu} W \sum_{\beta'(\Delta)}^{*} \varphi_{\Delta p, \Delta(p+2\mu), \Delta p}(0, \beta'; 1, \gamma)\Theta_{\Delta\left(p + 2\frac{\beta'}{\Delta}\right), \Delta p}(\tau, \overline{\tau}, z; 0, 1) \, . \tag{B.16}$$

We have already shown that the vectors in $\Lambda^*/\Lambda$ are exactly of the form $\beta'/\Delta$ where $\beta' \pmod{\Delta}$ is a special vector. Hence, the sum over $\beta'$ can be replaced by a sum over $\nu \in \Lambda^*/\Lambda$ with $\beta' = \Delta\nu$. Using (B.2) we can then also identify

$$(-1)^{\frac{1}{2}p^2 + p \cdot \mu}\Theta_{\Delta(p+2\nu), \Delta p}(\tau, \overline{\tau}, z; 0, 1) = (-1)^{p \cdot (\mu - \nu)}\Theta_\nu(\tau, \overline{\tau}, z) \, . \tag{B.17}$$

Using this and (A.36) we conclude that

$$
\begin{aligned}
\mathcal{O}(\gamma)\Theta_\mu(\tau, \overline{\tau}, z) &= W \sum_{\nu \in \Lambda^*/\Lambda} \varphi_{\Delta p, \Delta(p+2\mu), \Delta p}(0, \Delta\nu; 1, \gamma)(-1)^{p \cdot (\mu - \nu)}\Theta_\nu(\tau, \overline{\tau}, z) \\
&= W \sum_{\substack{\nu \in \Lambda^*/\Lambda \\ k \in \mathbb{Z}^n}} \sum_{k\,(c)} (-1)^{p \cdot k} e^{-\pi i \left[\frac{a}{c}(\mu + k + \frac{1}{2}p)^2 - \frac{2}{c}(\nu + \frac{1}{2}p) \cdot (\mu + k + \frac{1}{2}p) + \frac{d}{c}(\nu + \frac{1}{2}p)^2\right]} \\
&\quad \times (-1)^{p \cdot (\mu - \nu)}\Theta_\nu(\tau, \overline{\tau}, z) \, .
\end{aligned}
\tag{B.18}
$$

From (A.31) it follows that the factor $W$ is given by

$$W = \frac{1}{\sqrt{|\Delta|}} \left(\frac{-i(c\tau + d)}{c}\right)^{b_2/2 - 1/2} \left(\frac{i(c\overline{\tau} + d)}{c}\right)^{1/2} e^{-\pi i \frac{cz_-^2}{c\tau + d}} e^{-\pi i \frac{cz_+^2}{c\overline{\tau} + d}} \, , \tag{B.19}$$

where we have used that $b_- = b_2 - 1$ and $b_+ = 1$.

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
