# Peer review of "On the exact entropy of N = 2 black holes"

_SciPost Physics_

## Round 1 · Referee Report · Anonymous (Referee 1) · 2020-10-14

Strengths

1) Detailed analysis of the microscopic and macroscopic aspects of D4-D2-D0 black holes in N=2 string theory compactifications. 2) Derivation of the measure of Sen's quantum entropy functional for this class of black holes.

Report

This article studies the black hole entropy of D4-D2-D0 black holes and their M-theory duals. The authors study the partition function of this class of black holes and its relation to the 2004 conjecture by Ooguri, Strominger and Vafa, which relates black hole entropy to the topological string free energy. The authors further consider Sen's entropy functional and determine the measure in this path integral.

I find the article an interesting contribution, which will be useful in future work in this subject. I recommend it for publication by SciPost.

Requested changes

After reading the manuscript, I have a few comments and suggestions: 1) Eq. (2.6): Are the electric charges $q_a$ assumed to vanish, or should $q_0$ be $\hat q_0$? 2) Below (2.22): I suggest to add that $q_-^2$ and $q_+^2$ vanish both if and only if $q=0$. This is necessary for convergence. 3) Below (2.37): I find (ST)^3=S^2=-I. The elements I and -I are identical in PSL(2,Z), but I and -I act differently on the elliptic variable z of a Jacobi form, isn't it? 4) Above (2.54): I think a reference for the equivalence of D6-D4-D2-D0 partition function and topological string partition function will be helpful for the reader. 5) Below Eq. (2.66): My impression is that the lhs of (2.66) being a product formula is a consequence of the Schwinger calculation, but that the degeneracies $N_{q_,j_L,j_R}$ must be determined separately. 6) In (2.75): Should $\mathbb{R}$ be $\mathbb{R}^{b_2}$? 7) Eq. (2.96): The measure factor appears to resemble a symplectic inner product of the charges of the D6 and anti-D6 branes. Is this indeed the case? 8) Above (A.4): What are the special vectors $l$? 9) Appendices: I would suggest to edit the titles of Appendix A and B

---

## Round 1 · Referee Report · Anonymous (Referee 2) · 2020-11-3

Strengths

Proposed macroscopic computation of N=2 black hole entropy in string theory, a very challenging problem.

Weaknesses

The paper does not distinguish clearly results from conjectures, and some conjectures are rather suspicious.

Report

The authors study the exact degeneracy of N=2 supergravity BPS black holes in type IIA string theory. They derive a Rademacher expansion for the MSW BPS index that they compare to Denef and Moore refined OSV formula. They describe the Sen’s quantum entropy functional using localisation and an uplift to AdS2 x S1. Using several manipulations they claim that the two computations match up to some (very) conjectural identities.

String theory provides an explicite computation of the exact degeneracy for N=4 and N=8 BPS black holes in type II string theory. The situation is more complicated with only N=2 supersymmetry, in part because of wall-crossing and the difficulties to compute Donaldson-Thomas invariants. For N=4 and N=8 it has been shown that one can (up to some string theory inputs) derive the exact entropy using Sen’s quantum entropy functional and localisation techniques. The generalisation of the localisation techniques to N=2 supergravity is a very challenging and this is a very interesting problem in string theory. However, the results of the paper are very speculative and some of the conjectures are not really justified.

The paper starts with a short review of the MSW (4,0) SCFT. They write the BPS index as the scalar product of a vector valued Narain theta function and vector valued holomorphic modular forms. They give an explicit form of the vector representation matrices. They manipulate the Narain theta series to write the index as a Rademacher expansion, which they truncate to the first term c=1 for the rest of the paper. In section 2.3 they describe the refined OSV conjecture of Denef and Moore. They do not use the regularised partition function advocated by Denef and Moore, but take instead a somehow ad hoc truncation of a sum over positive integers in (2.68) that they further refine in the following by assuming convergence properties.

This truncation should be further justified. They say "It is interesting to see whether we obtain a similar truncation as in [15]" while they probably mean, "It would be interesting to check if this truncation is consistent with the prescription proposed in [15]". It is not clear how their truncation compares to [15], is does not seem to be either a stronger or a weaker truncation.

In (2.87) they explicitly find that their criterion is not enough to only get physical charges satisfying to (2.83). The conclusion that such charges do not satisfy to the extreme polar limit is not enough to find no contradiction, as they claim below (2.91). The positivity of the distance between bound-state constituents is a semi-classical formula derived in supergravity that can be generalised to any charges, so it should be satisfied. The use of the limit in reference [15] was to enforce that no further bound states, that are not considered by the authors either, can contribute.

The similar comment below (2.95) is not really encouraging. The sum over nu follows from the Rademacher expansion in (2.52), whereas the sum over m1 and m2 is not really justified. So it is not that it would be very interesting to compare, the authors should prove that they are equivalent up to negligible terms in the approximation scheme. Right after this sentence they conclude that the degeneracy (2.52) is equal to (2.77), whereas they just admitted they don’t understand the sum. I guess they mean that it is satisfied for a specific set of (m_i,n_i) that they have not precisely identified. Just below they write that one can only trust the formula for extreme polar states, which seems in contradiction with the problem found in 2.3, since their sum includes states that are not extreme polar from (2.91).

In Section 3 they discuss the macroscopic entropy using localization techniques. In (3.6) they give a solution with an unfixed \theta with respect to reference [24]. In reference [24] is is said that theta is simply a gauge choice, whereas in the following they need to fix theta to the precise value (3.29). How can it be that a gauge parameter has to be fixed to this specific value, shouldn’t it drop out of the computation? The argument using the Kahler potential in (3.32) is not very convincing, and it seems that the authors have simply fixed theta for their formula to work at next to leading order. Section 3.2.3 is an attempt to propose how instanton corrections could be included. Assuming the Ansatz (3.34), they find that they can reproduce their results of section 2 for a very strange looking prepotential (3.36), if they assume moreover that the prescription (3.32) for fixing theta should be followed. This is really too many if. The prepotential used in (3.10) is only an approximation, and the complete prepotential includes infinitely many higher order terms in both A and the Kahler moduli. Why the AdS2 instantons should dominate the pre-potential corrections? One of the authors himself comments on this difficulty in the first paragraph of page 4 in [58].

In general the paper does not distinguish clearly what is proved from what is conjectural and sometimes just wishful thinking. I believe there are some interesting results in the paper, but there are sections that should not be published as such. Section 3.2.3 in particular should not be published and sections 2.3.2 and 2.4 should be clarified. The generalisation of the localisation formula to N=2 supergravity is a very complicated problem, and I don’t think it would be shameful to restrict to the leading Bessel function in the Rademacher expansion. I think that the paper could be published in SciPost with important amendments along these lines.

Requested changes

1) Sections 2.3.2 and 2.4 should be clarified.

2) Section 3.2.3 should be removed.

3) More definitions should be explained. It would not arm, for example, to recall below (3.6) that A is the Weyl square chiral field and that its contribution in 3.10 gives rise to the Weyl square coupling in the effective action.

---

## Round 2 · Referee Report · Anonymous · 2022-7-1

Strengths
The authors have made progress in a very challenging problem in string theory.
Weaknesses
The paper does not distinguish clearly what is proved from what is conjectural.
Report
Let me first apologize to the authors for the extremely long delay before they could get a report. As the second referee I have only been asked to have a second look at the paper few months ago.
I acknowledge the fact that section 3.2.3 has been removed. However, the corrections in 2.3 and 2.4 are rather minimal, and do not clarify completely the situation.
In equation (2.68) the set C is still defined as the set of charges with m_a positive with respect to the d norm, whereas it is assumed everywhere as explained below (2.68) that they are moreover extreme polar. The authors change the definition such that there is no non-extreme polar states, but I do not understand what shows that this is the correct assumption. What I understand is that most of the analysis is valid for extreme polar states and the other cases remain to be analysed. In any case I believe the correct equation should be that they define C \subset and not equal to the set of discrete (n,m) with m positive such that the associated states are extreme polar. Talking about (n_i,m_i) bellow (2.68) is actually confusing since the pairs of charges only appear below in (2.69) when rewriting |ZGW|^2 as a sum of instantons.
The author say in the conclusion: we have mapped the Rademacher expansion to the degeneracy as derived by Denef and Moore, but I do not consider they have. They should put some conditional in this sentence, and resume what remains to be clarified.
I believe this paper is interesting and shed some light on the `macroscopic derivation’ of exact black hole degeneracy on N=2 string theory on a Calabi-Yau three-fold. However, I do not consider that the results are as strong as stated by the authors and they should explain clearly what has been proved and what is conjectural. The minimal modification of assuming C to only contain extreme polar states does not make the result more rigorous, whereas the conclusion only admits some points needed to be clarified regarding the supergravity computation. Therefore I will recommend the paper for publication if the authors accept to modify the paper accordingly.
Requested changes
1) (2.47) replace equal sign by approx or sim.
2) (2.52) replace equal sign by approx or sim.
3) (2.68) replace equal sign by subset. Replace (n_i,m_i) by (n,m).
4) State clearly in the conclusion what is proved and what is conjectural and requires further analysis. In particular explain the potential difference between their set C and the definition of Moore and Denef.

---

## Round 2 · Author Response

Thank you for sending us your detailed comments and recommendations in order to improve our work. In the list of changes, we will details how we included all these comments in the new version of our submission.
With kind regards,
Joao Gomes, Huibert het Lam and Grégoire Mathys

---

## Round 2 · List of Changes

1 Report 1: 2020-10-14
1. In Eq. (2.6) the electric charges $q_a$ are indeed assumed to vanish. In case of non-vanishing charges, $q_0$ should be replaced by $\hat{q}_0$. Since at this point in the paper we have not
introduced the electric charges yet, we have not mentioned this assumption explicitly.
2. We have added this as a footnote on page 8.
3. We adapted this part as well. We now work with SL(2; Z) instead of PSL(2; Z). This
implies that$ (ST)^3$ = $S^2$ = 1, which implies (2.38).
4. We have added appropriate references between equations (2.53) and (2.54).
5. The referee is correct. The degeneracies cannot be determined by a Schwinger type
computation. We have modied this in the text and added the appropriate citations.
6. The referee is right, and we changed the paper appropriately.
7. The referee is correct. The measure factor is indeed a symplectic product of the charges
of the D6 and anti-D6 branes, as can be seen using Eq. (4.26) in Denef(Moore (reference [15] of the paper).
8. The denition of a special vector is given below Eq. (A.1). Special vectors are integral
vectors x with n components such that Qx is divisible by detQ, i.e. each of its components
is divisible by detQ.
9. We changed the titles to proper English.
2 Report 2: 2020-11-3
1. We have adapted the paper, and stated from the beginning that we are just considering
extreme polar states in our truncation (Eq. (2.68)). This takes away some of the issues the
referee has brought up. For instance, this ensures that the distance between the branes
is always positive. Above (2.96) we indeed meant that this identication is satised for
a specic set of (mi, ni) that we have not precisely identied. However, we have derived
certain conditions that should be satised due to convergence properties. We comment
on having another domain for (mi, ni) in the last sentence of section 2.4.
2. We have removed section 3.2.3. We have also removed the subsequent section because
it was relying heavily on section 3.2.3. We have presented the idea that was contained
in section 3.2.3 in the discussion section. In addition, we have adapted all references to
these sections throughout the paper.
3. We have added the comment about the eld A^. We nevertheless don't think that any important
denitions are missing, but we would be happy to add any particular occurrences
that the referee would nd missing here.
With kind regards,
Jo~ao Gomes, Huibert het Lam and Gregoire Mathys

You are currently on this page

---

## Editorial Decision

unknown